



# Landsat NIR band and ELM-FATES sensitivity to forest disturbances and regrowth in the Central Amazon

Robinson I. Negrón-Juárez[1], Jennifer A. Holm[1], Boris Faybishenko[1], Daniel Magnabosco-Marra[2,3], Rosie A. Fisher[4,5], Jacquelyn K. Shuman[4], Alessandro C. de Araujo[6], William J. Riley[1], Jeffrey Q. Chambers[1]

[1] Lawrence Berkeley National Laboratory, Climate Sciences Department, 1 Cyclotron Road, Berkeley, CA 94720, USA.
[2] Max-Planck-Institute for Biogeochemistry, Hans-Knoell Str. 10, 07745 Jena, Germany.
[3] Brazil's National Institute for Research in Amazonia (INPA), Av André Araújo 2936, 690060-001, Manaus, Brazil
[4] National Center for Atmospheric Research (NCAR), 1850 Table Mesa Dr, Boulder, CO 80305, USA.
[5] Centre Européen Research et de Formation Avencée en Calcul Scientifique, (CERFACS) Toulouse, France
[6] Brazilian Agricultural Research Corporation -Embrapa) Eastern Amazon. Trav. Dr. Enéas Pinheiro, s/n°, Bairro Marco, CEP: 66095-903, Brazil

*Correspondence to*: Robinson I. Negrón-Juárez (robinson.inj@lbl.gov)

**Abstract.** Forest disturbance and regrowth are key processes in forest dynamics but detailed information of these processes is difficult to obtain in remote forests as the Amazon. We used chronosequences of Landsat satellite imagery to determine the sensitivity of surface reflectance from all spectral bands to windthrow, clearcutting, and burning and their successional pathways of forest regrowth in the Central Amazon. We also assess whether the forest demography model Functionally Assembled Terrestrial Ecosystem Simulator (FATES) implemented in the Energy Exascale Earth System Model (E3SM) Land Model (ELM), ELM-FATES, accurately represents the changes for windthrow and clearcut. The results show that all spectral bands from Landsat satellite were sensitive to the disturbances but after 3 to 6 years only the Near Infrared (NIR) band had significant changes associated with the successional pathways of forest regrowth for all the disturbances considered. In general, the NIR decreased immediately after disturbance, increased to maximum values with the establishment of pioneers and early-successional tree species, and then decreased slowly and almost linearly to pre-disturbance conditions with the dynamics of forest succession. Statistical methods predict that NIR will return to pre-disturbance values in about 39 years (consistent with observational data of biomass regrowth following windthrows), and 36 and 56 years for clearcut and burning. The NIR captured the observed successional pathways of forest regrowth after clearcut and burning that diverge through time. ELM-FATES predicted higher peaks of initial forest responses (e.g., biomass, stem density) after clearcuts than after windthrows, similar to the changes in NIR. However, ELM-FATES predicted a faster recovery of forest structure and canopy-coverage back to pre-disturbance conditions for windthrows compared to clearcuts. The similarity of ELM-FATES predictions of regrowth patterns after windthrow and clearcut to those of the NIR results suggest that the dynamics of forest regrowth for these disturbances are represented with appropriate fidelity within ELM-FATES and useful as a benchmarking tool.



**Keywords**: Landsat, disturbances, regrowth, Vegetation Demographic Models, Amazon

## 1 Introduction

Old-growth tropical forests are declining in extent at accelerated rates due to deforestation (Keenan et al., 2015), and they currently occupy an area about 50% of their original coverage (FAO, 2010). This decline affects the carbon, water, and nutrient cycles of the ecosystems and accelerates the loss of ecosystem goods and services (Foley et al., 2007;Nobre et al., 2016). Furthermore, natural and anthropogenic disturbances may act synergistically to exacerbate forest degradation (Silverio et al., 2018;Schwartz et al., 2017). Under natural conditions, disturbed forests recover to their pre-disturbance conditions through complex interactions that vary across spatial and temporal scales (Chazdon, 2014). In general, it is known that forest pathways of regrowth initiate with fast-growing and shade-intolerant species (pioneers) that establish from seeds and dominate a few years after disturbance, followed by recruitment and establishment of shade-tolerant species, and finally a closed-canopy old-growth forest (Chazdon, 2014;Denslow, 1980;Mesquita et al., 2001;Swaine and Whitmore, 1988). Understanding of the dynamics of forest regrowth following natural and anthropogenic disturbances in the Amazon, however, has so far been limited by lack of long-term observational data showing different stages of forest regrowth.

Remote sensing data can be used to assess forest regrowth via changes in spectral characteristics (Frolking et al., 2009;Roberts et al., 2004;Schroeder et al., 2011;DeVries et al., 2015;Lucas et al., 2002). Landsat satellite imagery is appropriate for examining land surface changes due to its long-term record availability and horizontal resolution of 30 m (Loveland and Dwyer, 2012;NASA, 2016;Wulder et al., 2012;Alcantara et al., 2011;Woodcock et al., 2008;Cohen and Goward, 2004;Hansen et al., 2013). Landsat imagery has been used to detect forest disturbance and pathways of regrowth in temperate and boreal forests in the Unites States and Canada (Kennedy et al., 2012;Pickell et al., 2016;Kennedy et al., 2007;Kennedy et al., 2010;Schroeder et al., 2011;Dolan et al., 2009;Dolan et al., 2017) and for detection of forest disturbance and regrowth of biomass in the Amazon (Vieira et al., 2003;DeVries et al., 2015;Lucas et al., 2002;Powell et al., 2010;Lu and Batistella, 2005;Steininger, 2000;Shimabukuro et al., 2019). These studies suggest that Landsat may be sensitive to different types of disturbances and their subsequent pathways of forest regrowth in the Amazon, but this has not yet been assessed.

The ability to forecast future trajectories of forests depends upon the fidelity with which disturbance and regrowth processes are represented within terrestrial biosphere models. These models capture processes operating between the leaf and landscape scales and can represent regrowth changes over large regions (Fisk, 2015), long time periods (Holm et al., 2017;Putz et al., 2014), a range of disturbance intensities (Powell et al., 2013), and interactions between multiple disturbance types and disturbance histories (Hurtt et al., 2006). But, an evaluation of how well these models simulate and capture the diverse array of successional pathways of forest regrowth after anthropogenic or natural disturbances needs to be more thoroughly evaluated,





given observed increases in disturbance rates (Lewis et al., 2015). The few modeling studies analyzing tropical disturbances have focused on the effects of fragmented edges or the regrowth of specific tree species (Dantas de Paula et al., 2015;Kammesheidt et al., 2002).

Cohort-based dynamic Vegetation Demographic Models (VDMs) are particularly suitable tools for expanding upon these studies (Fisher et al., 2018). In contrast to traditional land surface models, VDMs include ecological demographic processes, such as discretized vegetation height, with different plant types competing for light within the same vertical profile, and heterogeneity in light availability along disturbance and recovery trajectories, all of which facilitate direct simulation of regrowth dynamics during succession. This structured demography in VDMs allows for simulation of canopy gap formation,

competitive exclusion, and co-existence of vegetation, thus producing variability in forest stand age and composition (Fisher et al., 2010;Moorcroft et al., 2001;Longo et al., 2019). VDMs are designed for vegetation to dynamically respond to variation in traits (Fyllas et al., 2014) leading to differences in plant mortality, growth, and recruitment rates (Shugart and West, 1980). These attributes influence the ecosystem fluxes of carbon, energy, and water (Bonan, 2008). Despite their potential for simulating regrowth processes, there has been limited VDM testing of regrowth following tropical forest disturbances.

Importantly, projections of future climate using earth system models (ESMs) are strongly influenced by the terrestrial carbon cycle in the tropics (Arora et al., 2013;Friedlingstein et al., 2014), which is strongly regulated by disturbance and regrowth (Chazdon et al., 2016;Trumbore et al., 2015;Magnabosco Marra et al., 2018).

Observational studies have shown that Amazon forests follow a range of successional regrowth pathways after clearcutting

and burning (Mesquita et al., 2001;Mesquita et al., 2015). Thus, the type of disturbance and the pre-disturbance ecosystem state are important determinants of the successional pathways of forest regrowth. Nonetheless, this information is difficult to obtain in remote forests of the Amazon. In this study, we addressed this issue in the context of windthrow, clearcutting, and burning disturbances to analyze (*i*) the sensitivity of Landsat to detect and distinguish these relevant disturbances and their pathways of forest regrowth and (*ii*) the timespan of forest regrowth. This understanding of forest regrowth was used to (*iii*)

test the modeled forest regrowth of the Functionally Assembled Terrestrial Ecosystem Simulator (FATES) (Fisher et al., 2015) implemented in the Energy Exascale Earth System Model (E3SM) land model (ELM) (Riley et al., 2018): ELM-FATES. This study provides insights on the use of remote sensing to identify drivers of forest disturbance in the Amazon and a better understanding of the pathways of forest regrowth provides insights into the resilience of these forests to repeated disturbances and can help improve land models.


## 2 Study Area and Methods

### 2.1 Study area and sites





Forests in the Central Amazon affected by windthrow, clearcuting, and burning were addressed in this study. Windthrows (Mitchell, 2013) in the Amazon are caused by strong descending winds that uproot or break trees (Garstang et al., 1998;Negrón-Juárez et al., 2018;Nelson et al., 1994). In clearcut areas, forests are cut and cleared and in burning areas forest are cleared and burned (Mesquita et al., 2001;Mesquita et al., 2015;Lovejoy and Bierregaard, 1990). The windthrow, clearcut, and burned sites used in this study were selected based on the following conditions: (a) prior to disturbance they were upland old-growth forest

and located in the same geographic region, with no or minimal climatic, edaphic, and floristic differences; (b) long-term records of satellite imagery and corresponding field data before and after disturbance are available; and (c) no subsequent disturbance has occurred.

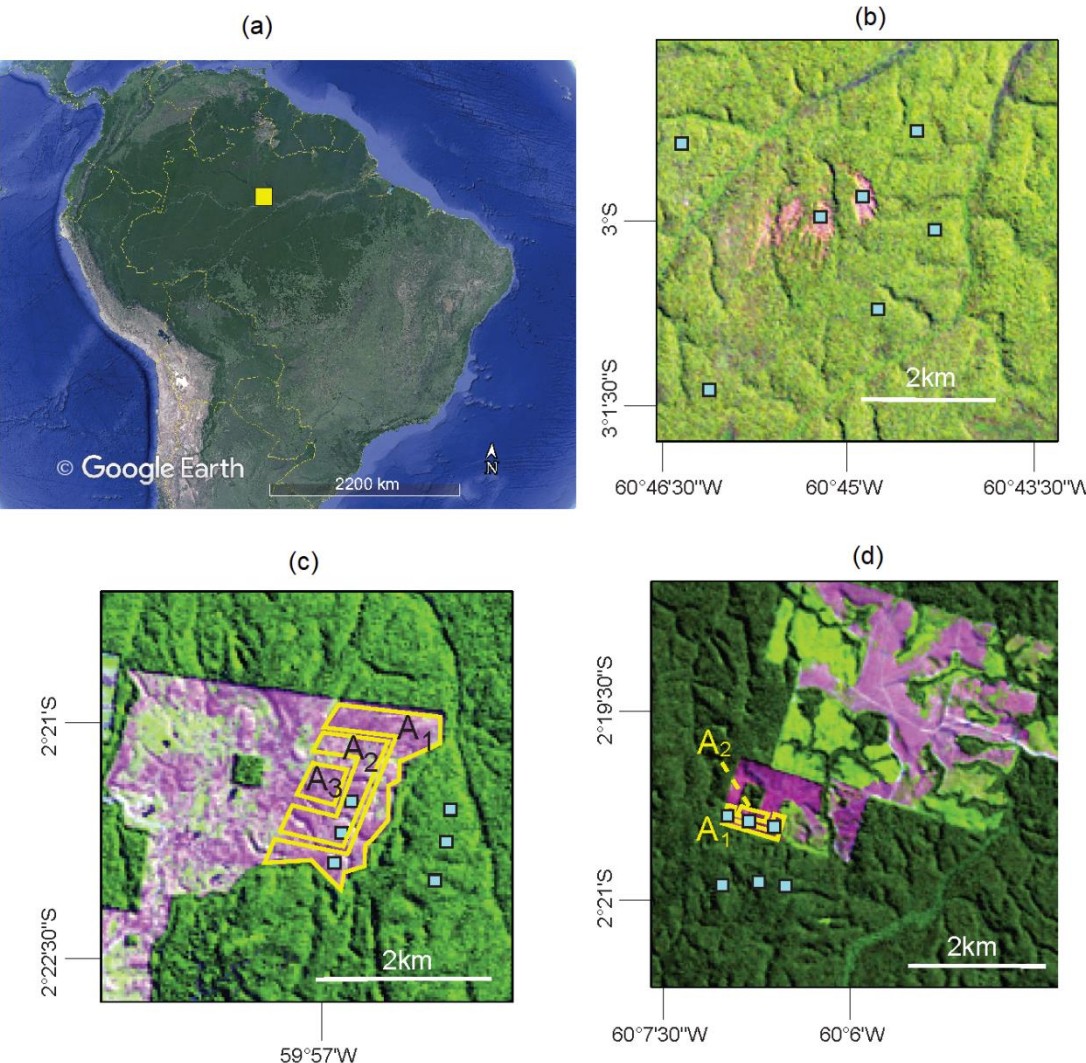



**Figure 1**: Location of disturbed forests. **(a) The disturbed areas were located in Central Amazonia and included (b) a windthrow site close to the village of Tumbira, (c) a clearcut site, in the Porto Alegre farm, and (d) a burned site in the Dimona farm. These three areas are encompassed in the Landsat scene Path 231 Row 062. For the spectral characteristics before and after disturbances we used boxes of 3×3 pixels (blue squares) over disturbed and undisturbed areas. For the pathways of forest regrowth after clearcutting and burning sites we analyzed different areas (A₁, A₂ and A₃ in yellow). The background image in (a) is from Google Earth Pro. The background images in (b), (c) and (d) are from Landsat 5 on July 12, 1987, June 1, 1984 and July 12, 1987 and were composed as RGB color using bands 5, band 4 and band 3, respectively.**

The windthrow, clearcut, and burned sites analyzed in this study are located near the city of Manaus, Central Amazon (Figure 1a). The windthrow (Figure 1b) was located near the village of Tumbira, about 80 km southwest of Manaus, occurred in 1987 and covered an area of ~75 ha. At this site, data on forest regrowth including forest structure and species composition for trees ≥10 cm DBH (diameter at breast height, 1.3 m) were collected since 2011 covering disturbed and undisturbed areas (Magnabosco Marra et al., 2018). The clearcut and burned sites were experimentally created within the Biological Dynamics of Forest Fragments Project (BDFFP), which encompass an area of ~1000 km² (centered at 2.5° S, 60°W) located 80 km north of the city of Manaus, Brazil. The BDFFP was established and managed in early 1980's by Brazil's National Institute for Research in Amazonia (INPA) and the Smithsonian Institution, and is the longest running experiment of forest fragmentation in the tropics (Bierregaard et al., 1992;Lovejoy et al., 1986;Laurance et al., 2011;Tollefson, 2013;Laurance et al., 2018). The selected clearcut site is located in the Porto Alegre farm (Figure 1c). This site was clearcut in 1982 without subsequent use, and was dominated by the pioneer tree genus *Cecropia* 6-10 years after abandonment (Mesquita et al., 1999;Mesquita et al., 2001). The burned site is located in the Dimona farm (Figure 1d), which was clearcut and burned in 1984 and maintained as pasture for some years and then abandoned. By 1993 this site was 6 years old and dominated by the pioneer tree genus *Vismia* (Mesquita et al., 1999;Mesquita et al., 2001).

In the Manaus region the mean annual temperature is 27°C (with higher temperatures from August to November, and peak in October) and the mean annual rainfall is 2,365 mm with the dry season (rainfall < 100 mm month$^{-1}$ (Sombroek, 2001)) from July to September (Negrón-Juárez et al., 2017). The topography is relatively flat with landforms ranging from 50-105 m above sea level (Laurance et al., 2011;Renno et al., 2008;Laurance et al., 2007), and the mean canopy height is ~ 30 m, with emergent trees reaching 55 m (Laurance et al., 2011;Lima et al., 2007). The soil and floristic composition in this region are summarized in Negrón-Juárez et al. (2017). In the BDFFP areas there are 261±18 species per hectare and 608 ± 52 stems ha$^{-1}$ (trees ≥ 10 cm in DBH) (Laurance et al., 2010). This stem density is representative of the region (da Silva et al., 2002;Vieira et al., 2004). In this region 93% of stems are between 10 and 40 cm in DBH (Higuchi et al., 2012) and the annual tree mortality is of 8.7 trees ha$^{-1}$ for trees ≥ 10 cm in DBH (Higuchi et al., 1997).



## 2.2 Landsat satellite data and procedures


The Landsat Ecosystem Disturbance Adaptive Processing System (LEDAPS) (Schmidt et al., 2013;Masek et al., 2006;Masek et al., 2013;Masek et al., 2008) surface reflectance (SR) from Landsat 5 Thematic Mapper (TM) was used in this study to characterize the type of disturbance and their subsequent pathways of forest regrowth over our study areas. LEDAPS was developed to ensure that spectral changes in Landsat are associated with regrowth dynamics (Masek et al., 2012;Schmidt et

al., 2013) and to facilitate robust studies of land surface changes at different temporal and spatial scales in tropical forests (Kim et al., 2014;Valencia et al., 2016;Alonzo et al., 2016). LEDAPS SR Landsat 5 TM (L5 hereinafter) is generated by the United States Geological Survey (USGS) using the Second Simulation of a Satellite Signal in the Solar Spectrum (6S) that corrects for the influences of, among others, water vapor, ozone, aerosol optical thickness, and digital elevation on spectral bands (USGS, 2017;Vermote et al., 1997). L5 are derived using per-pixel solar illumination angles and generated at 30-meter spatial

resolution on a Universal Transverse Mercator (UTM) mapping grid (USGS, 2017). LEDAPS in Landsat 7 ETM+ sensor (L7, hereinafter) were also used to corroborate our predictions (described below). Though L5 and L7 use the same wavelength bands they are different sensors and differences in surface reflectance may exist, especially in tropical forests (Claverie et al., 2015). Landsat 8 was not used since comparison between Landsat 8 and both L5 and L7 is not straightforward due to differences in the spectral bandwidth of these sensors. We used LEDAPS since has a long time series of data, is promptly

available, have high spectral performance comparable to Moderate Resolution Imaging Spectroradiometer (MODIS) (Claverie et al., 2015) and several datasets (Vuolo et al., 2015;Nazeer et al., 2014) and it is suitable for ecological studies in the Amazon (van Doninck and Tuomisto, 2018;Valencia et al., 2016). L5 and L7 are available in Google Earth Engine (Gorelick et al., 2017), which we used to retrieve and analyze these data.

The L5 and L7 spectral bands used in this study were BLUE (0.45-0.52μm), GREEN (0.52-0.62 μm), RED (0.63-0.69 μm), Near Infrared (NIR) (0.76-0.90 μm), Shortwave Infrared 1 (SWIR1) (1.55-1.75 μm), and Shortwave Infrared 2 (SWIR2) (2.08-2.35 μm). L5 and L7 measurements provide the fraction of energy reflected by the surface and ranges from 0 (0%) to 10000 (100%). Only scenes from June, July, and August were used since these months present less cloud cover over our study area (Negrón-Juárez et al., 2017). This procedure also reduces effects associated with illumination or phenology since images

correspond to the same period each year. Only images with cloud free, cloud shadow free, and haze free over our disturbed areas were used to eliminate errors associated with these elements. For this procedure, visual inspection of visible bands and quality information from L5 and L7 were used. No further corrections were applied due to the robustness of L5 imagery over the Amazon (Valencia et al., 2016). The dates of L5 images used were (Landsat 5 operational imaging ended in 2011) 6/1/1984, 7/6/1985, 7/12/1987, 8/2/1989, 7/20/1990, 8/8/1991, 7/31/1994, 6/21/1997, 7/26/1998, 7/13/1999, 7/24/2003,

8/4/2007, 8/6/2008, 7/27/2010, and 8/31/2011. The dates of L7 images used were 8/7/2011, 6/22/2012, 6/12/2014, 8/2/2015 and 8/7/2017.



The spectral characteristics of old-growth forests and their changes after disturbances were investigated using several boxes of 3×3 pixels (Figure 1 b,c,d) as shown in Figures 3a-c. Spectral characteristics for old-growth forest for each site were
determined from boxes located in the same position of the disturbance but previous to disturbance, as well as from adjacent areas. Five boxes of old-growth forests were located from 1 to 2 km away from the windthrow site. Though closer distances may also represent old-growth forests, we were conservative since Landsat is not sensitive to clusters of downed trees comprising fewer than 8 trees (Negrón-Juárez et al., 2011). For the clearcut and burned sites the spectral characteristics of their respective old-growth forests were studied from 3×3 boxes located 500 to 800 meters away from the edge of the disturbance
to minimize edge effects that are relevant in the first 100 m (Lovejoy et al., 1986;Laurance et al., 2007;Mesquita et al., 1999). Boxes containing old-growth forests were also used as controls. The spectral characteristics for the windthrow were acquired from two boxes containing the highest level of SWIR1 that is associated with the maximum disturbance (Negrón-Juárez et al., 2011;Magnabosco Marra et al., 2018). For the clearcut site three boxes were located 400-500 m from the edge and for the burning three other boxes distant from 100-300 m with respect the edge. The spectral characteristics of old-growth and
disturbances are shown is Figure 3 d-f with the error bands representing the standard deviation of all pixels from respective boxes.

L5 data for the windthrow, clearcut, and burned sites encompass a period of 27 years with 12 years of missing data due to cloud-cover or lack of image. In order to assess the forest regrowth to spectral levels similar to old-growth forests (control),
we applied a gap filling method (Gerber, 2018) of time series to obtain estimates for missing years using the R package "zoo" (Zeileis et al., 2018). The gap-filled datasets were analyzed using the smoothing spline technique (R, 2017).

To determine whether L5 bands were sensitive to regrowth, we analyzed changes in the slope ($\beta$) of the bands across our chronosequence. A t-test on the slope coefficient was used to test the null hypotheses that $\beta$ is zero ($H_0:\beta=0$) against the
alternative hypothesis ($H_1:\beta\neq0$) at a 5% significant level ($\alpha=0.05$). If the computed test statistic (t-stat) was inside the critical values then the $H_0$ was not rejected. The critical values ($\pm t_{1-\alpha/2, n-2}$, n is the number of data points) were obtained from statistical tables (Neter et al., 1988). Forests affected by windthrows are dominated by tree species from genera *Cecropia* and *Pourouma* in about 3-5 years (Magnabosco Marra et al., 2018;Nelson and Amaral, 1994) and the clearcut and burned sites were dominated by *Cecropia* and *Vismia* about 6 years after the disturbances (Mesquita et al., 1999;Mesquita et al., 2001). The slopes of the
time series were determined after these periods, i.e. 1991, 1987, and 1990 for windthrow, clearcut, and burned sites, respectively.

A comparison of successional pathways of forest regrowth among studied disturbances was conducted that was possible due to the similar conditions of climate, soils, structure, and composition of the old-growth forests. For windthrows, we analyzed
the areas of maximum disturbance since the spectral signature of these areas last the longest in Landsat imagery (Nelson et al.,





1994). For clearcut and burned sites we used areas with different distances from the disturbance edge to determine whether distance is a factor affecting the regrowth. For the clearcut site we selected four areas: $A_1$, $A_2$, $A_3$, and $A_T$ ($A_T = A_1+A_2+A_3$), shown in Figure 1c and for the burned site, three areas: $A_1$, $A_2$, and $A_T$ ($A_T =A_1+A_2$), shown in Figure 1d. L5 bands were analyzed using the statistical nonparametric function (univariate fit), with the smoothing spline and the Gaussian regression

ANOVA (analysis of variance) model. Calculations were conducted on the R 3.5.2 software platform (R, 2017) using the package gss (general smoothing splines) (Gu, 2018). We calculated the smooth spline (using the cubic fit algorithm) of observed data and the associated standard errors, from which we calculated Bayesian 95% confidence intervals. Predictions of the time after disturbance needed to reach old-growth forests values are based on these data using the function "ssanova" (Fitting Smoothing Spline ANOVA Models) of the R package "gss" (General Smoothing Splines), Version: 2.1-9. The

predictions were compared with field observations where data were available and L7 images were used to assess the reliability of our predictions.

## 2.3 Forest regrowth simulation in ELM-FATES

Time series of L5 bands sensitive to disturbances and the pathways of forest regrowth were compared with modeling results from FATES (Fisher et al., 2015;Fisher et al., 2010). The underlying model structure and concepts in FATES are based on the Ecosystem Demography (ED) concept (Moorcroft et al., 2001), and is described in detail at https://github.com/NGEET/fates. A major development is the modularization of the model structure in FATES so that boundary conditions and vegetation can be coupled with ESM land models. FATES is integrated into the E3SM Land Model (ELM) (Riley et al., 2018;Zhu et al.,

2019) and within the Community Land Model (CLM) (Fisher et al., 2019;Wieder et al., 2019) coupled to the Community Earth System Model (Hurrell et al., 2013). In this study we used ELM-FATES. ELM-FATES simulates vegetation that varies in successional age and size, plant competition, and dynamic rates of plant mortality, growth, and recruitment, all on landscapes partitioned by areas of disturbance. The main updates and modifications in ELM-FATES compared with ED include changes to carbon allocation and allometry and introduction of the Perfect Plasticity Approximation (PPA) (Purves et al., 2008;Fisher

et al., 2010) used for the accounting of crown spatial arrangements throughout the canopy and organizing cohorts into discrete canopy layers. Photosynthesis and gas exchange physiology in ELM-FATES follows the physics within the Community Land Model v4.5, CLM, (Bonan et al., 2011), and unlike ED, uses the original Arrhenius equation from Farquhar et al. (1980). ELM-FATES tropical forest simulations conducted here were based on parameter and demography sensitivity analysis at a site close to BDFFP (Holm et al., 2019). Holm et al. (2017) found that with the improved parameterization, ELM-FATES

closely matched observed values of basal area, leaf area index (LAI), and mortality rates but underestimated stem density for a Central Amazon old-growth forest near the BDFFP.

Model simulations were driven by climate-forcing data derived from measurements collected between the years 2000 to 2008 at the K34 flux tower located at (2.6°S, 60.2°W) (de Araujo et al., 2002) about 40 km from the BDFPP. ELM-FATES (using



the git commit "4a5d626" and the version corresponding to tag 'sci.1.0.0_api.1.0.0') was run and spun-up for 400 years until a stable biomass equilibrium in the model was reached within the forest. We then simulated a one-time logging treatment of a near-complete mortality of all trees (98% "clearcut") with a remaining of 2% consisting of only small trees > 5cm DBH to aid in recruitment (Mesquita et al., 2015). Second, we simulated a one-time windthrow disturbance that killed 70% of trees ("windthrow") as was reported in a recent observational study on windthrows in central Amazon (Magnabosco Marra et al.,

2018). All dead trees were 'removed' and therefore did not enter modeled soil pools. Burning module in ELM-FATES is currently under tests and therefore burned simulations will not be done in this study. The old-growth forest simulated by ELM-FATES, used as a pre-disturbance metric, was based on previously validated tropical parameterization and sensitivity testing (Holm et al., 2019); and see supplemental material from Fisher et al. (2015) for description of plant functional type specific carbon allocation and allometry schemes and updates from the ED model framework. Simulations of vegetation regrowth after

disturbance were initiated from this old-growth forest state. Field data was not used to simulate or calibrate the modeled forest recovery post disturbance.

To account for uncertainty in the representation of plant physiology within tropical evergreen forests, we analyzed an ensemble of 20 simulations varying in targeted plant functional traits. We prescribed each ensemble with a single tropical evergreen

plant functional type (PFT) that varied in wood density (0.44 to 1.06 g cm$^{-3}$) and maximum rate of carboxylation ($V_{cmax}$; 42 to 55 umol m$^2$ s$^{-1}$) (Table 1), via random sampling. To evaluate changes in canopy coverage of the forest stand each PFT additionally varied by an allometric coefficient (1.35 to 1.65) determining crown area to diameter ratio, and a leaf clumping index (0.59 to 1.0 out of 0-1 fraction) that determines how much leaf self-occlusion occurs and decreases light interception, and the direct and diffuse extinction coefficients in the canopy radiation calculations. The default values for these parameters

are based on, or derived from references given in Table 1. Each ensemble member represents a single PFT across the spectrum of fast-growing 'pioneer' PFTs and slow-growing 'late successional' PFTs to provide a reasonable spread across the trait uncertainty when assessing regrowth from disturbance. We characterized pioneer plants in our simulations as having low wood density (Baker et al., 2004) and high $V_{cmax}$ based on the inverse relationship between these two plant traits, as well as a low crown area coefficient and low leaf clumping factor; i.e., monolayer planophile distribution (Lucas et al., 2002). These

correlated relationships were applied in the ensemble-selected traits (Figure 2). The opposite relationship was applied for slow-growing, 'late successional' PFTs (e.g., high wood density, low $V_{cmax}$, high crown area coefficient, and high leaf clumping factor).

**Table 1. The range (minimum to maximum) of four key model input parameters used in the 20-ensembles ELM-FATES**
**simulations for both windthrow and clear-cut simulations, to account for uncertainty in the representation of plant traits, along with the default value used in the ELM-FATES model. Wood density value from Moorcroft et. al. (2001), $V_{cmax}$ based on Oleson et al. (2013) and Walker et al. (2014), crown area:DBH derived from Farrior et al. (2016) and**




adjusted based on site specific sensitivity tests, and the leaf clumping index based on radiation transfer theory of
Norman (1979).


| | Variations in ensemble parameters in ELM-FATES | | | |
| --- | --- | --- | --- | --- |
| | default | minimum | maximum | Range |
| Wood density (g cm$^{-3}$) | 0.7 | 0.44 | 1.06 | 0.62 |
| Vcmax (μmol m$^2$ s$^{-1}$) | 50 | 42 | 55 | 13 |
| Crown area : DBH (unitless) | 1.5 | 1.35 | 1.65 | 0.30 |
| Leaf clumping (0-1) | 0.85 | 0.59 | 1.00 | 0.41 |

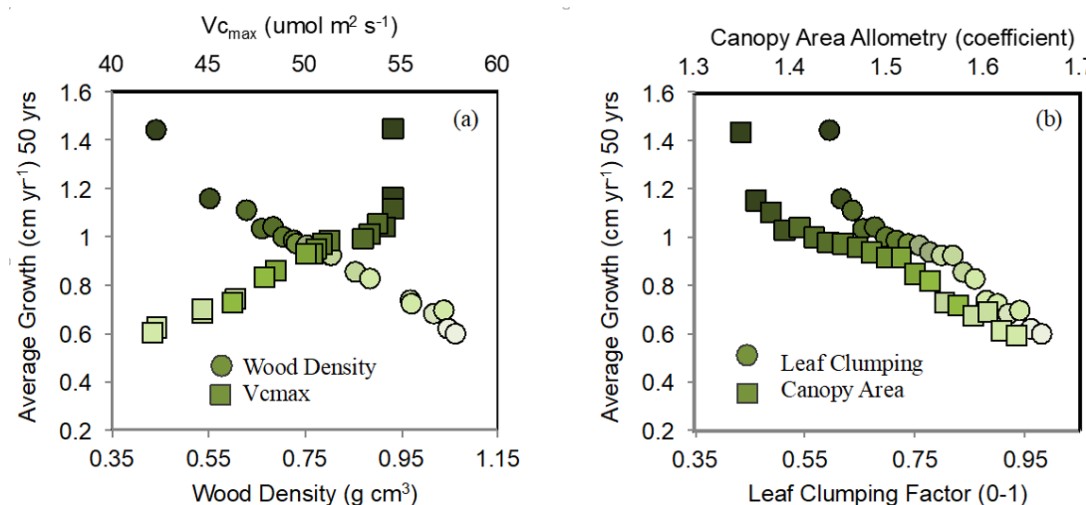

Figure 2: Imposed trait variation used in the parameterization of ELM-FATES tropical evergreen plant functional
types (PFTs) for the 20-ensemble simulations and the resulting average growth rate average over the 50-year simulation
period. Each simulation consisted of a single PFTs varying by all four traits at once: wood density, Vc$_{max}$, the canopy
area allometric coefficient, and leaf clumping index for leaf self-occlusion. Dark green points represent fast-growing
evergreen pioneer PFTs, while light green points represent slow-growing late successional PFTs.

In order to evaluate ELM-FATES performance during forest regrowth we compared NIR with ELM-FATES outputs of
aboveground biomass (AGB, Mg ha$^{-1}$), total stem density of trees ≥10 cm DBH (stems ha$^{-1}$), leaf area index (LAI, one-sided
green leaf area per unit ground surface area, m$^2$ m$^{-2}$), and total crown area (m$^2$ m$^{-2}$) since these variables directly influence the
surface reflectance (Ganguly et al., 2012;Lu, 2005;Masek et al., 2006;Powell et al., 2010;Ruiz et al., 2005). We suggest that
including an array of forest variables (e.g., biomass structure, density coverage of vegetation, and proportion of the tree crown
that has live foliage) for comparison to NIR provides a robust comparison across several variables that affect NIR reflectance



and reduces unknowns and biases when using only one model variable. The usage of different stand structure and canopy processes can be helpful when evaluating ELM-FATES during different phases of forest regrowth. In addition, we averaged outputs of crown area, stem density, and LAI since each of these variables influence the surface reflectance, and defining this average as the modeled 'canopy-coverage'. Measurements of forest canopy cover have been used to analyze plant growth and

survival, and it is an important ecological parameter related to many vegetation patterns (Ganey and Block, 1994;Jennings et al., 1999;Paletto and Tosi, 2009). Modeled diameter growth (cm y$^{-1}$) once trees were ≥10 cm at 1.3 m was also show to provide results on the successional dynamics of forest stands typical of early or late successional characteristics within a demographic model.

**3 Results**

**3.1 L5 bands and disturbances**

Overall, all L5 bands showed an increase in surface reflectance associated with windthrow, clearcut, and burned sites except

NIR which decreased (with higher decrease after burning) (Figure 3 a, b and c). This decrease in NIR was due to exposed woody material and dry leaves (typical after windthrow and clearcutting) or the dark surface following burning. Such effects last about one year after which vegetation regrowth covers the ground surface (Negrón-Juárez et al., 2010a;Negrón-Juárez et al., 2011). About one year after the disturbance the bands that experienced increase in surface reflectance showed a decrease in surface reflectance (the opposite for NIR) (Figure 3d - f) due to the increases in vegetation cover. A similar behavior is

expected for clearcutting that occurred in 1982 and therefore before the beginning of our available data (L5 imagery are available from 1984, Figure 3e). On the other hand, in our control (old-growth) forests, we observed typically high NIR reflectance due to the cellular structure of leaves (Chapter 7 in Adams and Gillespie, 2006), absorption of red radiation by chlorophyll (Tucker, 1979), and absorption of SWIR1 by the water content in leaves (Chapter 7 in Adams and Gillespie, 2006). The similarity of spectral signatures for the control forests previous to the disturbances suggests comparable structure and

species composition.

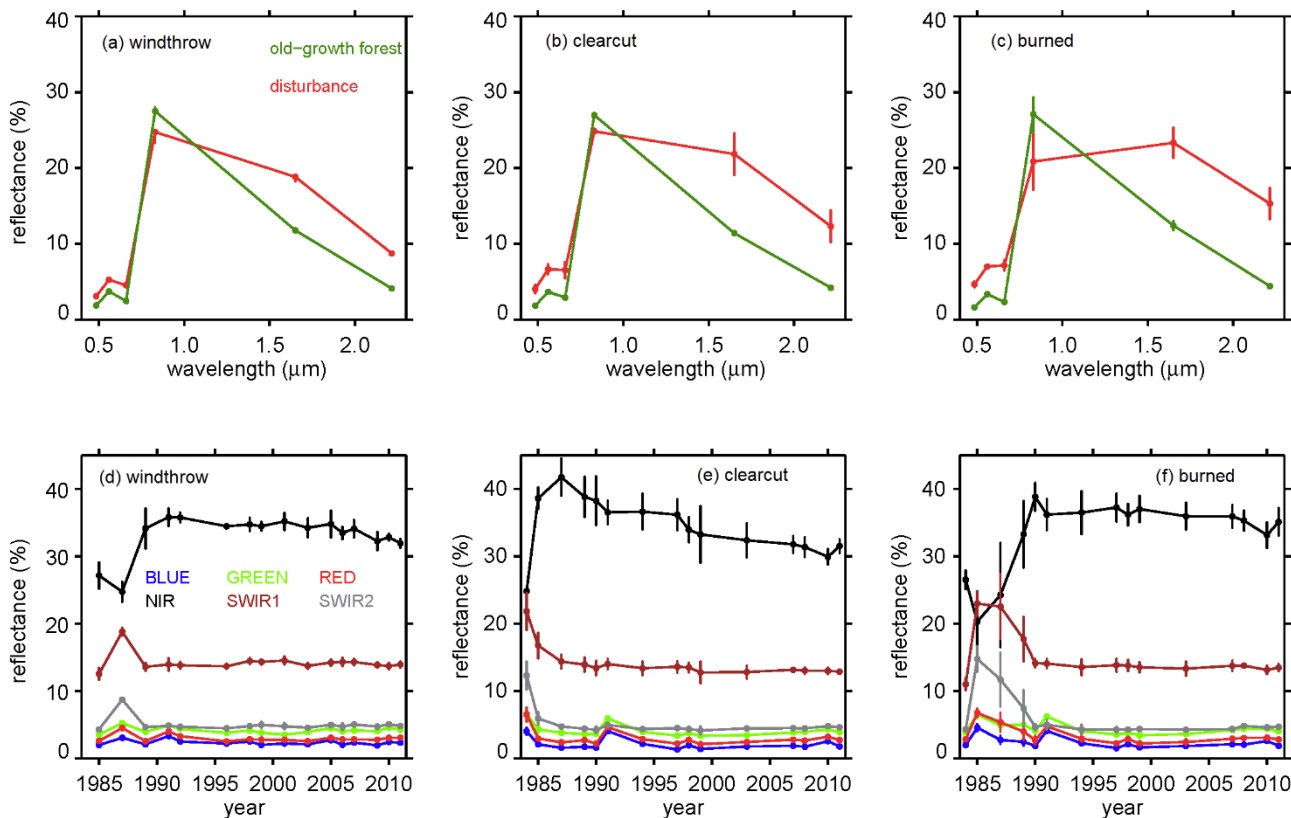

**Figure 3: L5 (LEDAPS SR Landsat 5) spectral characteristics for (a) windthrow (July 12, 1987), (b) clearcut (June 1 , 1984), and (c) burned (July 12, 1987) (in red) and control (old-growth) forests (in green) sites. Time series of each L5 spectral bands for (d) windthrow, (e) clearcut, and (f) burned sites. The bars represent the standard deviation from all pixels from all 3×3 boxes comprising the respective disturbances showed in Fig. 1.**

## 3.2 Pathways of forest regrowth

About six years after the disturbance, NIR reached a maximum value after which it decreased slowly with time showing a significant negative trend (Table 2). SWIR1 also showed a significant negative trend with time but only for the clearcut site (Table 2). In general, GREEN, BLUE, RED, SWIR1, and SWIR2 bands returned to pre-disturbance values (control) about six years after the disturbance (Figure 3d, e, f and Table 2). Therefore, we used NIR (which remained higher than pre-disturbance values throughout the time series, and is potentially sensitive to ecosystem properties of re-growing forest) to investigate the regrowth dynamics in comparison to our control forests.





We used the relationships presented in Figures 4, 5, and 6 to determine the time that NIR from the disturbance sites became similar to control NIR. The average control NIR was 28±1%. For the windthrow site the NIR become similar to control levels after about 39 years (range 32 to 57 years). For the clearcut and burned sites, this period was estimated to be 36 years (range

to 42 years) and 56 years (range 42 to 93 years), respectively. From Figure 4-6 it is evident that the type of disturbance has a clear effect on the pathways of NIR recovery. L7 data, in general, are within the 95% CI of predictions.

**Table 2. Test of the significance for the slopes of the time series of six bands from L5 (LEDAPS SR Landsat 5)**
**windthrow, clearcutting, and burning cases in Central Amazonia shown in Figures 3d-f. The critical values ($t_{0.975,8}$ and**
**$t_{0.975,12}$) for the $t$ distribution were obtained from statistical tables. Bolt represents H1.**

| | windthrow $t_{0.975,12}$=2.179 | | clearcut $t_{0.975,8}$=2.306 | | burned $t_{0.975,8}$=2.306 | |
|---|---|---|---|---|---|---|
| | β | t-stat | β | t-stat | β | t-stat |
| BLUE | -1.51 | -1.10 | -0.63 | -0.25 | -3.92 | -1.15 |
| GREEN | -0.03 | -0.02 | -0.72 | -0.31 | -3.03 | -0.78 |
| RED | -0.12 | -0.68 | -0.18 | -0.08 | -3.19 | -0.93 |
| NIR | **-12.36** | **-4.07** | **-35.1** | **-10.17** | **-11.72** | **-2.83** |
| SWIR1 | 0.87 | 0.70 | **-4.83** | **-4.52** | -2.25 | -1.98 |
| SWIR2 | 0.95 | 1.45 | 0.17 | 0.22 | -0.48 | 0.36 |


During the first 12 years following the windthrow, the spline curve fitted to the NIR data decreased by ~0.13% y$^{-1}$ after which the rate of decrease doubled (Figure 4). For clearcutting, NIR decreased faster, i.e. ~0.4% y$^{-1}$. The decrease of NIR for the clearcutting site appears to be independent of the distance from the edge of the disturbance since the changes in NIR of all

selected areas (A$_1$, A$_2$, A$_3$ and A$_T$) are similar (Figure 5). For the burned site, the rate of change of NIR to values similar to the control forests was the slowest among all disturbances considered (~0.15%) (Figure 6). The burned site showed differences with respect to the border of the disturbance (areas A$_1$, A$_2$, and A$_T$), which may be related to the spatial heterogeneity of burnings and forest responses.





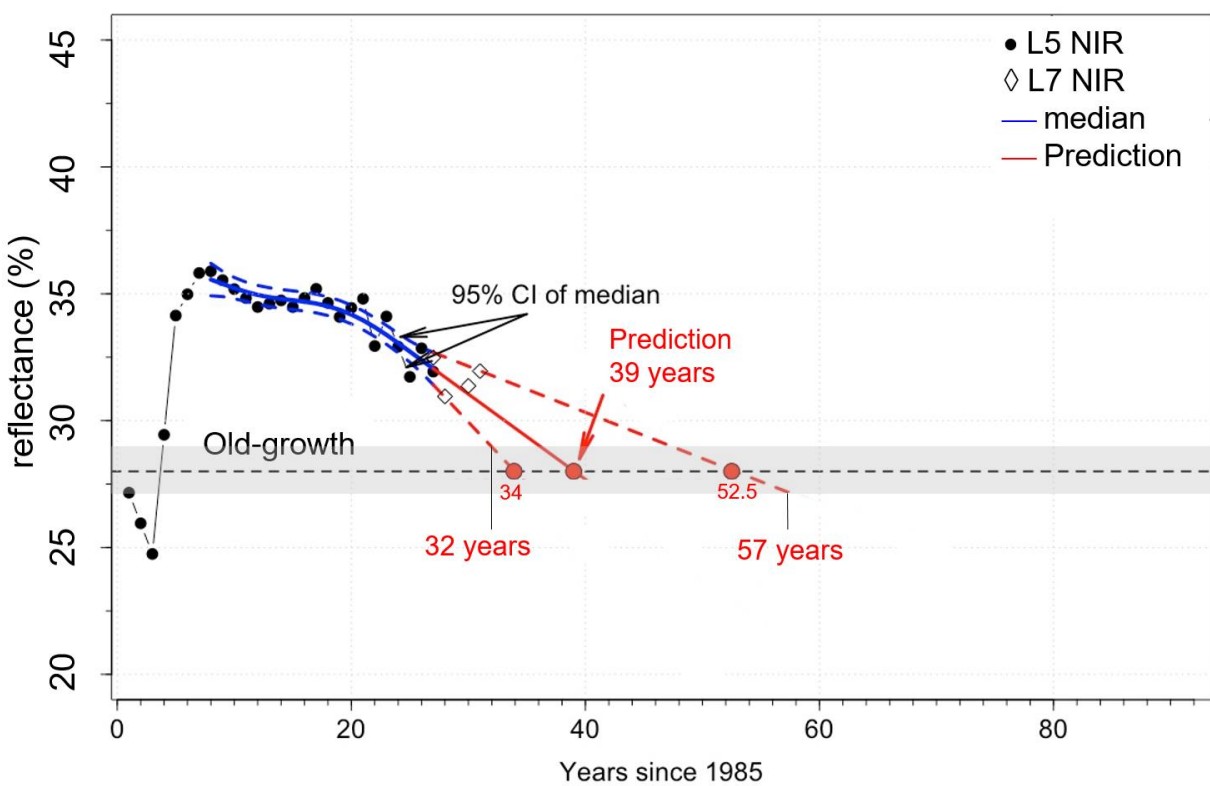


**Figure 4: Changes in NIR for windthrows and prediction (based on extrapolated of fitted spline curve) of NIR to pre-disturbance values (in blue). The plots show the SR data (dots), the fit (solid lines), and the 95% confidential interval (CI, dashed lines). Grey bar represents the control (old-growth forests) NIR of 28±1% and the black horizontal dashed line is 28%.**

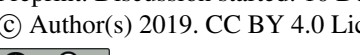


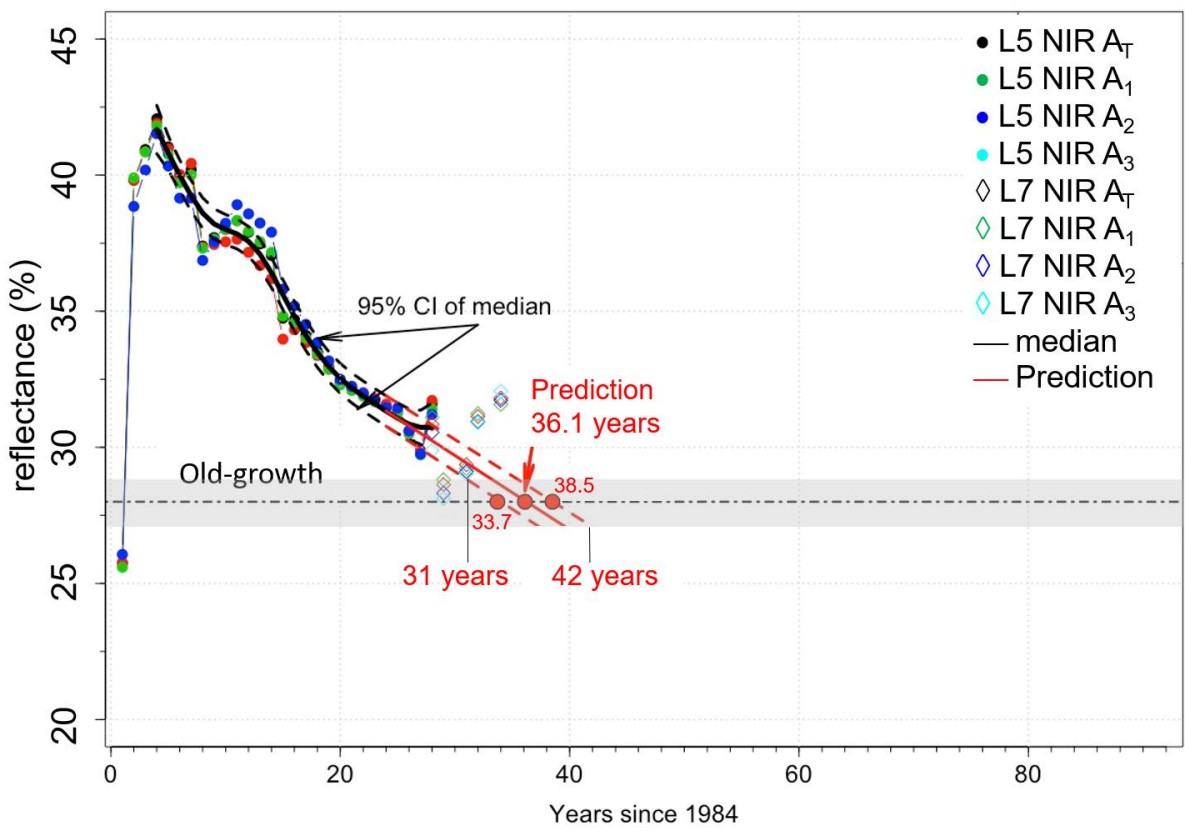


**Figure 5: Changes in NIR after clearcuts for areas $A_1$, $A_2$, $A_3$, and $A_T = A_1 + A_2 + A_3$ (showed in Figure 1c) and prediction of NIR to pre-disturbance values (in blue). The plots show the data (circles), the fit (solid line), and the 95% confidential interval (CI, dashed lines). Grey bar represents the control (old-growth forests) NIR of 28±1% and the black horizontal dashed line is 28%.**



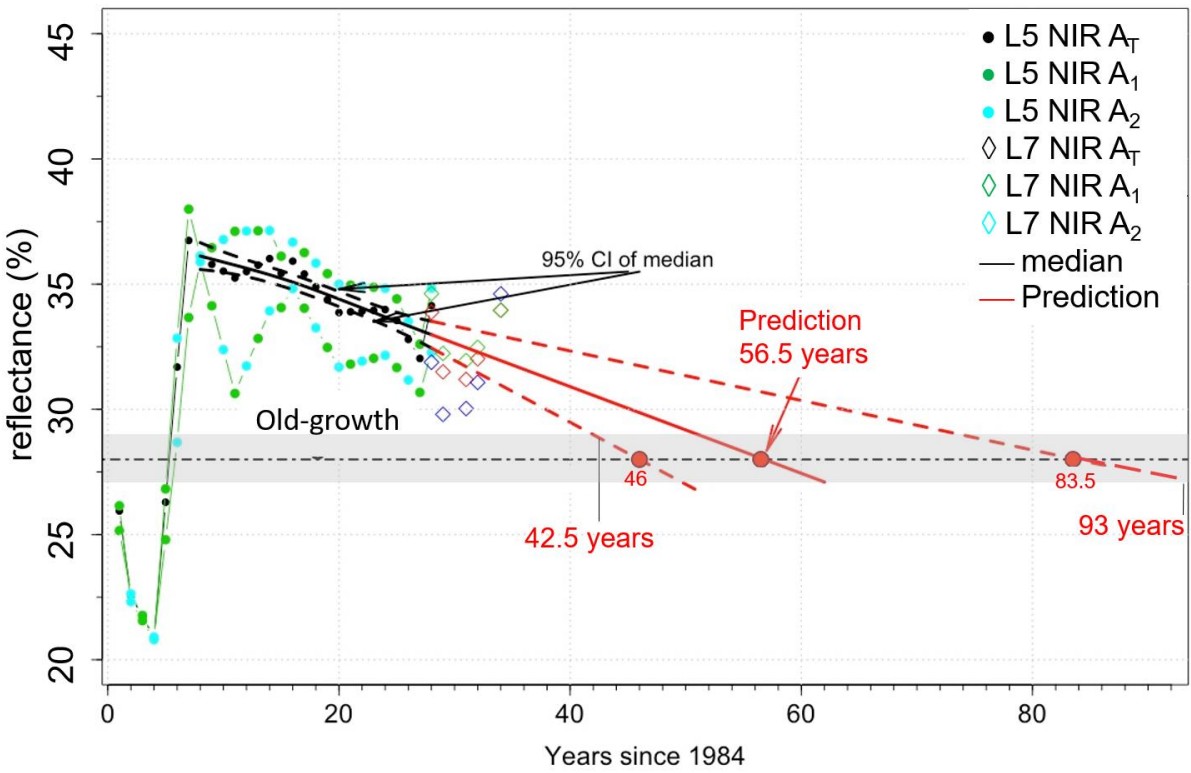


**Figure 6: Changes in NIR for burned site in areas A₁, A₂ and Aₜ =A₁+A₂ showed in Figure 1) and prediction of NIR to pre-disturbance values (blue). The linear fit (solid liner) and the 95% CI (dashed line) are shown. Grey bar represents the control (old-growth forests) NIR of 28±1% and the black horizontal dashed line is 28%.**


### 3.3 FATES model and regrowth from forest disturbance

To address our goal of improving the connection between remote sensing and model benchmarking and the fidelity of future predictions of forest regrowth processes, we examine the representation of such processes within ELM-FATES. The average

of the ELM-FATES 20-member ensemble predicted a continuous, and almost linear, regrowth of biomass (Figure 7a) after both clearcut and windthrows. Modeled biomass returned to modeled old-growth forest values quicker for windthrows (37 years, range 21 to 83 years) compared to clearcuts (42 years, range 27 to 80 years), with old-growth characterized as when biomass reached equilibrium with values similar to observed old-growth biomass. However, the rate of change of biomass



regrowth over 50 years was faster in the clearcut simulation (2.5 Mg ha$^{-1}$ yr$^{-1}$) than the windthrow simulations (2.0 Mg ha$^{-1}$ yr$^{-1}$), which is likely due to the near-zero initial biomass and proportionally greater contribution of fast-growing pioneer species.

The model simulation of stem density, LAI and crown area are shown in Figures 7b-d, respectively. For stem density, ELM-FATES (black line) predicted an average of up to eight years before any new stems reached ≥10 cm DBH (a stand developing period, Figure 7b) for the clearcut. The simulated stem density for old-growth forests (Figure 7b; green line) was ~200 stems ha$^{-1}$ (≥10 cm at 1.3 m), ~ 408 trees lower than observations. The model ensembles with typical early successional traits predicted a forest with many fast-growing, small-diameter stems <10 cm, with maximum early successional stem densities reaching 1,560 and 1,414 stems ha$^{-1}$ for clearcut and windthrows, respectively. Once the canopy closes and self-thinning dominates (average of 15 years after disturbance), there are fast declines in stem density as trees gain more biomass, and canopy closure forces some trees into the understory, where they die at faster rates due to shading. The model predicts faster diameter growth increments (1.3 cm yr$^{-1}$) and canopy closure for a forest comprised of all 'pioneer' type PFTs and slower (0.5 cm yr$^{-1}$) diameter growth and more open canopies for 'late successional' forest type (Figure 8). Diameter growth is an emergent model feature of dynamical plant competition for light and stand structure demographics, and in coherence with observational studies of secondary forests growing through succession (Brown and Lugo, 1990;Winter and Lovelock, 1999;Chapin III et al., 2003). The modeled forests return to old-growth stem density conditions (i.e., 200 stems ha$^{-1}$) after 39 and 41 years from windthrows and clearcut, respectively (Table 3). Thought they were very similar, they differ in the speed of recovery (Figure 7), as discussed below.

Due to disturbance, the initial modeled LAI (Figure 7c) and total crown area (Figure 7d) decreased, as expected. During regrowth from disturbance both LAI and total crown area rapidly recovered and surpassed pre-disturbance values. This behavior resembles the initial spike in NIR due to fast growing PFTs. These two canopy coverage attributes reached maximum values after 3 to 6 years, depending on the disturbance and canopy attribute (Table 3). To evaluate model results against remote sensing observations, we compared the initial period after the disturbance of the spikes in NIR to the 'canopy-coverage' metric (combination of LAI, stem density, total crown area) over the same modeled period. ELM-FATES predicted that after a windthrow the forest took 5.7 years to reach maximum values of canopy-coverage, which was sooner than the clearcut simulation (7 years). While the modeled timespan for this initial period was similar to the NIR, there was disagreement between which disturbance recovery occurred fastest (windthrow in ELM-FATES vs. clearcut in NIR), similar to disagreement in recovery of AGB (Table 3).



**Figure 7: Simulated regrowth of the Central Amazon forest after a clearcut (98% tree mortality; black line), and windthrow event (70% tree mortality; orange line) using 20 simulations of the demographic model ELM-FATES, compared to the modeled old-growth values prior to disturbance (green line). The shaded grey and orange areas represent the spread across the ensembles, showing minimum and maximum values of each forest attribute over its regrowth. (a) Regrowth of aboveground biomass (AGB; Mg C ha$^{-1}$). (b) Regrowth of stem density (stems ha$^{-1}$) of stems >10 cm DBH and years of returning to modeled old-growth values. (c) Regrowth of leaf area index (m$^{-2}$ m$^{-2}$), and (d) regrowth of total crown area (m$^{-2}$ m$^{-2}$).**





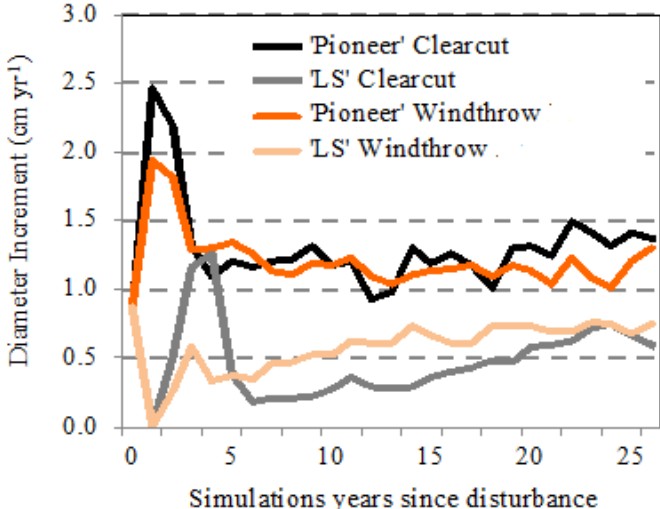

**Figure 8: Change in predicted diameter increment growth rate (cm yr⁻¹) for one simulation, from the 20 ensembles,**
**that represented a fast-growing 'pioneer' forest stand and a slow-growing 'late successional' (LS) forest stand from a**
**clear-cut disturbance (black and grey) and a windthrow disturbance (orange). Variations in diameter increment are a**
**result of differences in the following traits: wood density, $Vc_{max}$, crown area, and a leaf clumping index.**


**Table 3. Summary of different times of regrowth (years) to old-growth forest status after two disturbance types;**
**windthrows and clearcuts from ELM-FATES model results and remote sensing. As well as the time (years) it takes**
**forest attributes to reach maximum values during regrowth, and the corresponding value at this maximum peak. AGB**
**(Mg C ha⁻¹), stem density (stems ha⁻¹), LAI, and crown are (m² m⁻²) refer to simulation results, as compared against**
**NIR remote sensing. The average of AGB and stem density is characterized as modeled 'forest structure'. The average**
**of crown area, stem density, and LAI is characterized as modeled 'canopy-coverage' in this study, and additionally**
**compared against NIR.**







| Regrowth to old-growth (years) | | | | |
|---|---|---|---|---|
| Disturbance type | ELM-FATES AGB | ELM-FATES stem density | ELM-FATES LAI | NIR | Model average of forest structure |
| Windthrow | 37 | 39 | 53 | 39 | 38.0 |
| Clearcut | 42 | 41 | 26 | 36.1 | 41.5 |
| Time to reach maximum values of regrowth (years) | | | | | |
| Disturbance type | ELM-FATES crown area | ELM-FATES stem density | ELM-FATES LAI | NIR | Model average of canopy-coverage |
| Windthrow | 5 | 9 | 3 | 7 | 5.67 |
| Clearcut | 5 | 10 | 6 | 6 | 7.0 |
| Values at maximum peak of regrowth | | | | | |
| Disturbance type | ELM-FATES crown area ($m^2$ $m^{-2}$) | ELM-FATES stem density (stems $ha^{-1}$) | ELM-FATES LAI ($m^2$ $m^{-2}$) | NIR (%) | |
| Windthrow | 0.99 | 1414 | 4.9 | 35.5 | |
| Clearcut | 0.98 | 1560 | 7.5 | 42.0 | |


ELM-FATES provided a prediction of the values in each forest variable when the stand reached its production limit and full canopy closure, at which point there was a shift to a declining trend and decreases in forest attributes that outpaced any gains (Table 3). The highest values, at peak recovery and carrying capacity limit, occurred in the clearcut simulation (with a very small exception in crown area), matching the higher NIR from clearcut some years after the disturbance (Figure 4-6). Over the

longer self-thinning period modeled LAI decreased and returned to modeled old-growth values 26 years after clearcut, and gradually over 53 years for windthrows (Table 3). LAI was the only variable that had a faster recovery in the clearcut simulations. After both disturbances the total crown area permanently remained higher (0.65 $m^2$ $m^{-2}$) than the crown area of the simulated old-growth forests (0.58 $m^2$ $m^{-2}$), suggesting that disturbances generates a denser canopy, as discussed below.

**4. Discussion**

Our results show that Landsat reflectance observations were sensitive to changes of vegetation regrowth following windthrows, clearcut, and burning, three common disturbances in the Amazon. Specifically, a decrease in NIR and an increase in SWIR1 were the predominant spectral changes immediately (within a few years) following disturbances. The increase in SWIR1 was

different among the disturbances with the maximum increase observed in the burned, followed by clearcut and then the windthrow site. The highest increase in SWIR1 in burned sites may be related to the highest thermal emission of burned





vegetation (Riebeek, 2014). Likewise, the relatively higher moisture content of woody material in the windthrow site decreases the reflection of SWIR2.

While SWIR1 is frequently used to identify exposed woody biomass immediately after disturbances (Negrón-Juárez et al., 2010b;Negrón-Juárez et al., 2008), we found that NIR was more sensitive to the successional pathways of regrowth for all the disturbances considered. NIR has also been associated with succession (Lu and Batistella, 2005) and regrowth (Roberts et al., 1998) in natural and anthropogenic disturbed tropical forests (Laurance, 2002;Chazdon, 2014;Magnabosco Marra, 2016;Laurance et al., 2011). Maximum values of NIR were observed about 6 years after clear cut, which is the time pioneers

form a closed canopy in the Amazon (Mesquita et al., 1999;Mesquita et al., 2015;Mesquita et al., 2001). NIR found that this maximum was higher in the clearcut site dominated by species from the genus *Cecropia* and *Pourouma* (Mesquita et al., 2015;Massoca et al., 2012) than the site affected by burnings dominated by *Vismia* species (Mesquita et al., 2015;Laurance et al., 2018). The higher NIR values in *Cecropia* and *Pourouma* is due to their monolayer planophile distribution of large leaves that produced high reflectance different from *Vismia* that have a more rough and dense canopy that traps more NIR (Lucas et

al., 2002).

After the establishment of pioneers, the NIR decreases with time but with different rates depending on the type of disturbances. In windthrown areas, tree mortality and subsequent recruitment may continue for several decades, promoting changes in functional composition and canopy architecture (Magnabosco Marra et al., 2018). *Cecropia* and *Pourouma* trees grow

relatively quickly and after closing the canopy they limit light penetration due to their large leaves creating a dark, cooler, and wetter understory (Mesquita et al., 2001;Jakovac et al., 2014). As a result, light levels in the understory decline faster with time and thus allow the recruitment and establishment of shade tolerant species. The cohort of *Cecropia* and *Pourouma* species has relatively short lifespan and ~20 years after disturbance, secondary and old-growth forest species start to establish (Mesquita et al., 2015). Consequently, light is intercepted by several understory layers/strata formed by different guilds of

trees, which drives decreases in NIR over time. With the self-thinning of *Cecropia* and *Pourouma*, the growing understory traps more light and consequently albedo decreases (Roberts et al., 2004). This pattern is consistent with the decline of NIR and observed changes in canopy architecture (Mesquita et al., 2015), photosynthesis, and LAI (Saldarriaga and Luxmoore, 1991). In contrast, the architecture of *Vismia* species that dominate burned areas allows higher light levels in the understory and subsequent recruitment of *Vismia* or other genera with similar light requirements. As a consequence, species turnover and

structural changes are relatively slower than in clearcutting (as Jakovac et al. (2014) found in a study conducted a few kilometers from the BDFFP) and windthrows, consistent with changes in NIR. In the course of succession, *Vismia* tends to be replaced by *Bellucia*, which is a species with similar leaf and canopy structure as *Vismia*. This pattern favors the penetration of light through the canopy (Longworth et al., 2014) for several decades before a more shaded understory allows the germination and establishment of old-growth species (Williamson et al., 2014).




For the windthrow we estimate that the NIR should become similar to pre-disturbance conditions in about 39 years. This value is in agreement with ground-based estimates of biomass regrowth from windthrows in the Central Amazon (40 years) (Magnabosco Marra et al., 2018). This result corroborates previous studies that NIR operates in the best spectral region to distinguish vegetation biomass (Tucker, 1979, 1980). For clearcutting and burning, the regrowth time was about 36 and 56

years respectively, but no ground-based estimates were available for comparison. Still, NIR showed that the pathways of regrowth from clearcut and burning are divergent with time (Figure 5 and Figure 6), which is consistent with observational studies (Mesquita et al., 2015).

In general, we found that NIR may be used as a proxy in modeling studies aimed at addressing forest regrowth after

disturbances. Though NIR is useful to distinguish successional stages up to decades after the disturbance, it may not represent the whole successional processes. As soon as the forest canopy becomes structurally similar to that of the mature forest, NIR will no longer be sensitive to changes in vegetation attributes (Lucas et al., 2002). Thought L5 NIR may be complemented with current Landsat measurements - L5 NIR has comparable performance to the Landsat 8 NIR Operational Land Imager algorithm (OLI) (Vermote et al., 2016)-, it is important to emphasize that our estimates of recovered reflectance and biomass

in disturbed areas do not capture succession in floristic and functional composition. A full recovery of diversity in floristic attributes and species composition will take much longer. Furthermore, the predominance of *Cecropia,* after clearcut, and *Vismia,* after burns, have also been found in the Western (Gorchov et al., 1993;Saldarriaga et al., 1986) and the Southern (Rocha et al., 2016) Amazon suggesting that our findings are applicable to other regions. However, an Amazon-wide study is beyond the scope of our work which is to explore the sensitivity of Landsat to differentiate disturbance types.


Our analysis demonstrates that this version of ELM-FATES has the capacity to reproduce initial response to disturbance and regrowth patterns after the clearcut and windthrow that occurred over similar time ranges compared to NIR. The strongest agreement, which can be used for future benchmarking, occurred because ELM-FATES predicted higher peaks of initial stem density and LAI in clearcuts than in windthrows, consistent with the higher peak of NIR from clearcuts (Figure 5 vs. Figure

4). This effect may be due to ELM-FATES having more homogeneous canopies after clearcuts as well as more open disturbed area for fast growing plants, which is also an observed trend. In addition, the average regrowth times to pre-disturbance values were very similar in ELM-FATES and NIR (Table 3), showing that pathways of forest regrowth in ELM-FATES are comparable to observed patterns in tropical forests. ELM-FATES predicted a continuous, and almost linear, regrowth of biomass for the first 50 years of simulation after both clearcut and windthrows, (Figure 7a) consistent with NIR results, and

observational studies (Mesquita et al., 2015;Saldarriaga et al., 1988;Jakovac et al., 2014;Magnabosco Marra et al., 2018). In addition the changes in biomass rates predicted by ELM-FATES were similar to biomass observations recorded after clearcut (2.3 vs. 2.6 Mg ha$^{-1}$ yr$^{-1}$) (Mazzei et al., 2010), as well as a faster rate of AGB accumulation after clearcut compared to windthrow, similar to a study reporting higher regrowth rates in more highly-disturbed sites (Magnabosco Marra et al., 2018).



Landsat showed a faster recovery of NIR to pre-disturbance conditions in clearcuts compared to windthrows. Faster growth is characteristic of anthropogenically driven secondary forests that reflect rapid colonization and consequent monodominance of highly adapted species and genera to changed environmental conditions; e.g., high growth rates, low self-competition, high leaf area index, low herbivory rates, etc. (Poorter et al., 2016;Mesquita et al., 2015;Rozendaal and Chazdon, 2015). Alternatively, ELM-FATES predicted a faster recovery of structural AGB and canopy-coverage to pre-disturbance conditions

for windthrows (70% tree mortality) compared to clearcuts (98% tree mortality). A major contributing factor to this pattern resulted from larger modeled diameter increments after windthrows (0.92 cm yr$^{-1}$) compared to clearcuts (0.82 cm yr$^{-1}$) in the first 20 years after the disturbance, setting the trajectory for faster regrowth to pre-disturbance after windthrows. Only LAI had a faster recovery to pre-disturbance values after clearcuts, which is expected due to the newly developed forest having simplified forest structure and the canopy being more homogeneous after a clearcut (Rosenvald and Lohmus, 2008). ELM-

FATES predicted that the timing of peak canopy-coverage was marginally sooner after windthrows compared to clearcuts, opposite to the NIR pattern. This discrepancy may be related to the higher disturbance in the modeled clearcut. Due to the higher stand disturbance that naturally occurs from clearcuts, and the diverse complexities in tropical forest composition, we emphasize that the dynamics of different competing PFTs in ELM-FATES requires further investigation. Emerging modeling studies that include plant trait trade-offs, for example, in leaf and stem economic spectrum or fast vs. slow growth strategies

may help to better capture the drivers of forest productivity and demography, enabling improved modeled responses to global change scenarios (Fauset et al., 2019;Sakschewski et al., 2015). Here we test the basic representation of biomass demographics, prior to the more challenging aspects of representing interacting functional diversity in recovering systems (Fisher et al., 2015;Powell et al., 2018).

ELM-FATES predicts the total stem density for a closed canopy forest to be very low compared to observations (200 simulated vs 600 stems ha$^{-1}$), but modeled total AGB was close to that reported for the same region (110 simulated vs. ~150 MgC ha$^{-1}$ (Chambers et al., 2013)). This discrepancies is due to ELM-FATES predicting a disproportionately high number of large trees (Figure 9; with 8% of stems >60 cm and 4.5% of stems >100 cm), resulting in a crowded canopy, which out-compete smaller understory trees. Higuchi et al. (2012) reports 93% of stems to be below 40 cm (and ≥ 10 cm) in DBH in the study sites, while

ELM-FATES only has 88% below 40 cm (Figure 9). Low stem density could be attributed to multiple model assumptions, such as high density-dependent mortality and self-thinning due to the marginal carbon economics of understory trees, low branch-fall turnover, a need for greater limitation of maximum crown area than currently modeled, and increases in mortality rates with tree size (Johnson et al., 2018). Our findings here will guide future ELM-FATES and ecosystem modeling development efforts towards improving the representation of forests comprised of dense canopies, and how they shift during

regrowth.

Land surface models do not typically simulate spectral leaf reflectance, but there is potential to include such output within radiative transfer schemes as is currently done in CLM. That addition would greatly assist our ability to compare with Earth

observations datasets. In lieu of this development, we show that with successional aging modeled forest structure returns to

pre-disturbed values with similar recovery time as NIR, occurring with the process of canopy closure, all which can be

compared against remote sensing metrics.

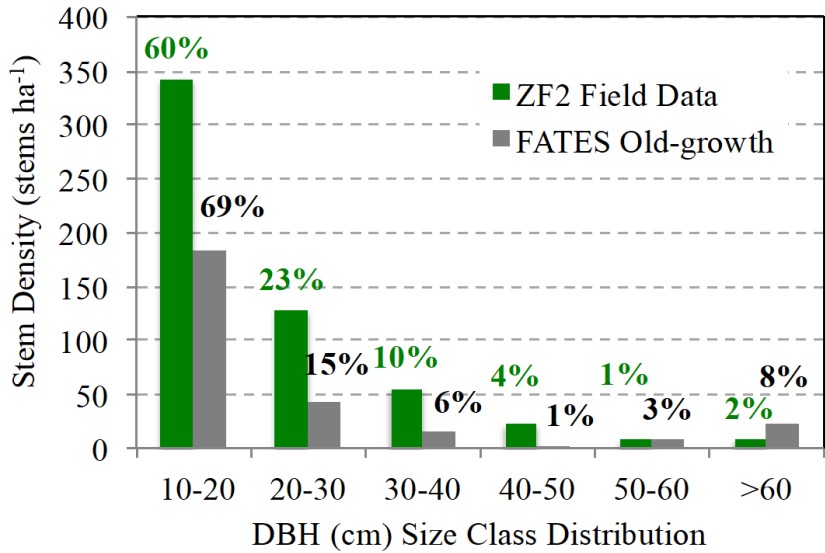

**Figure 9: Total stem density (stems ha⁻¹) separated into six diameter (cm) size classes from Central Amazon field data located at the near-by ZF2 site (green bars) averaged from 1996-2011, and predicted by ELM-FATES (gray bars). The percentages represent the proportion of stems in each size class relative to the total stem density.**

**5. Conclusions**

We tested the sensitivity of Landsat surface reflectance to windthrow, clearcut, and burned forest in Central Amazon. NIR was more responsive to the successional pathways of forest regrowth years after the disturbance. NIR showed that pathways of forest regrowth were different among the disturbances, with burning being the most different in terms of spatial heterogeneity

and regrowth time to old-growth status, in agreement with observational studies. Our results indicate that after disturbances the NIR will reach old-growth forests values in about 39 years following windthrows (in agreement with observed biomass regrowth), 36 years for clearcutting, and 56 years for burning. These results were then compared with simulations of regrowth after windthrows and clearcut from ELM-FATES. The simulated forest structure and the remote sensing NIR from the windthrow and clearcut have similar return time to old-growth forest conditions. Future studies applying ELM-FATES should

focus on improving stem density predictions, which were underestimated, and on enhancing the capacity to compare with remote sensing observations through representation of canopy spectral reflectance characteristics.

## 6. Acknowledgments

This research was supported as part of the Next Generation Ecosystem Experiments-Tropics (NGEE-Tropics) of the TES Program and the RUBISCO project under the Regional and Global Climate Modeling Program, funded by the U.S. Department of Energy, Office of Science, Office of Biological and Environmental Research under contract DE-AC02-05CH11231. Among others, the NGEE-Tropics Program supports a large part of past and current model development for the ELM-Functionally Assembled Terrestrial Ecosystem Simulator (FATES). DMM was supported as part of the ATTO Project, Max Planck Society (MPG) and the German Federal Ministry of Education and Research (BMBF). RAF was supported by the National Center for Atmospheric Research, which is sponsored by the National Science Foundation under Cooperative Agreement No. 1852977.

## 7. Code and Data Availability

The Landsat data used in this study is freely available through the Google Earth Engine platform. ELM-FATES is available at https://github.com/NGEET/fates. Observational data used to compare remote sensing and modeling results have been previously published and references provided.

## 8. Author Contribution

Robinson Negrón-Juárez designed the study and made the remote sensing analysis. Jennifer A. Holm made the model simulations and analysis. RNJ and JAH formulated the research goals and wrote the initial draft of the manuscript. Boris Faybishenko performed the statistical analysis of the remote sensing data. Daniel Magnabosco-Marra provided the field data for windthrows and contributed with the discussion of biomass recovery. Rosie A. Fisher and Jacquelyn K. Shuman contributed with modeling improvements. Alessandro C. de Araujo contributed with the observational input data to initialize the model. William J. Riley and Jeffrey Q. Chambers were responsible for project funding and administration. All authors contributed with the final writing and edition of the manuscript.

## 8. Competing Interests: No

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
