# Peer review of "Landsat NIR band and ELM-FATES sensitivity to forest disturbances and regrowth in the Central Amazon"

_Biogeosciences, 2019_

## Referee Comment (RC1) · Anonymous Referee #1 · 5 Jan 2020

Remotely-sensed forest disturbance and change monitoring has been proposed as a basis for improving and constraining vegetation modeling frameworks (McDowell et al. 2015). The current research (Negrón-Juárez et al. in review) proposes and tests components of such a framework by examining the utility of Landsat time series (LTS) data for detecting forest disturbance and recovery dynamics and comparing these results to ELM-FATES predictions for several forest attributes. Using LTS data to identify disturbance is common, but assessing recovery trajectories and timing of recovery is much less common. The use of LTS data for assessing forest dynamics processes is still a relatively new area of research (Schroeder et al. 2007). More rarely still has been the use of insights into recovery dynamics to assess the capacity of vegetation models

to reflect reality and, perhaps, guide model development. Such efforts are particularly challenging in the cloud-covered tropics. The primary conclusions – Landsat-based NIR observations are sensitive to forest regrowth dynamics which compare favorably to model predictions – may prove useful both for the development of monitoring frameworks and for guiding vegetation model testing and development in tropical forests.

The paper is well-structured, well-written, and of appropriate length for the material. The citations seems generally appropriate, though I do identify additional literature that should be discussed in the paper (see below). The title and abstract provide a clear and concise description of the work. I do suggest that the authors consider revising the final sentence of the abstract to provide a more impactful conclusion reflecting the potential impact of this work on disturbance and recovery mapping as well as vegetation modeling research. The methods and assumptions seem appropriate, though additional justification of some methods is required (I enumerate those in more detailed comments below). It should be noted that the small sample size of locations used in this study limits the generality of conclusions, but does seem sufficient to assess whether the proposed framework for integrating remote sensing with ELM-FATES is useful. The analysis was straightforward and seems reproducible by other scientists.

Here I raise several issues that should be addressed to improve the quality and accessibility of the paper.

1. I found the lack of use, or even discussion, of commonly used spectral vegetation indices, such as NBR, NDVI, EVI, or tasseled-cap wetness, greenness, and brightness, to be a significant oversight. Spectral vegetation indices have seen extensive use in remote sensing of terrestrial vegetation (Bannari et al. 1995), including disturbance ecology. For example, NBR is a common basis for disturbance severity mapping (e.g., Key and Benson 2006; Miller and Thode 2007). The signal-to-noise ratio for disturbance mapping in North America tends to be greater for spectral vegetation indices compared to Landsat spectral bands (Cohen et al. 2018). Furthermore, vegetation indices have proven useful in forest biomass mapping (e.g., Foody et al 2003) and forest

regrowth monitoring (e.g., Schroeder et al. 2007). I understand that these metric need not form the basis for the current study, but recognizing their application elsewhere and their potential to contribute to future research would be valuable. In particular, some discussion of common spectral vegetation indices that include NIR and/or SWIR1 in the context of the observed sensitivity to regrowth would be valuable.

2. In lines 130-132, it is stated that the burned sites were clearcut, then burned, then maintained as pasture for a few years before forest regrowth began. I am concerned with referring to cut and then burned areas simply as "burned sites". Fire as a mortality/disturbance agent has a different impact than fire used as post-disturbance vegetation management. Fire as a disturbance agent within forests will produce various levels of fire severity (e.g., Alves et al. 2018), and thus tree mortality, whereas post-disturbance fire is likely used to benefit forage species and remove woody vegetation, not kill trees in the forest canopy. The latter point is implied by the use of these sites as pasture for several years. It would be more appropriate to refer to these as "cut+burn" or "cut/burn" or something like that. The fire and/or grazing seems to have impacted species composition of the regrowing forest, but attributing the dynamics observed solely to fire seems inappropriate. Furthermore, since no burn simulations were used, is there any reason to have the burned sites without referring to the harvesting history as well? It is likely more appropriate and more interesting to reframe them as cut and burned, which is what they are based on the text. Then, the authors have two types of harvesting, which perhaps provides a more robust assessment of the ELM-FATES model and the complexity of management activities that should be incorporated in the modeling.

Even with these concerns, I found the paper to be an interesting attempt to leverage remote sensing in the testing of ecosystem models. Similar applications could lay the groundwork for new developments in Earth system modeling, especially in regions that are traditionally data poor, such as the Amazon.

**Specific and Technical Comments**

Line 16: Enumerate which sensors (TM and ETM+) or missions (Landsat 5 and 7) provided the data.

Line 42: Forest or successional pathways need to be defined early. Are the authors referring to succession of structural or compositional characteristics?

Lines 114-115: Why the different areas? What distinguishes them? I see them defined in lines 211-213. Perhaps the figure caption could reference the distance to edge component of these areas?

Lines 159-162: I think that there may be a few missing words in here.

Lines 178-179: Was each pixel in the 3x3 pixel box treated as independent (n=9) or averaged (n=1)?

Line 179: Figure 3 is cited earlier in the text than figure 2. Consider swapping them or delete the reference to Figure 3 (which I don't think is necessary)

Lines 193-196: This paragraph seems out of place. As I understand it, the gap-filled data is used in the analysis of the forest regrowth timing described later (lines 213-221). It would be easier if the description of the gap-filling methods appeared just before the analysis of those data.

Lines 208-209: I was confused by this sentence. Should it read "The comparison . . . disturbances that was conducted was possible due. . .". Still, I don't agree fully with the sentiment that controlling for those know environmental gradients makes the analysis possible. Perhaps the authors mean that by controlling for these other factors, they make the assessment of forest successional pathways following disturbance more robust.

Lines 209-211: Do the authors mean that greater magnitude wind disturbances tend to have longer recovery times?

Lines 254-256: Considering the fact that forest recovery is a major focus on this

manuscript, I suggest adding some more detail. Specifically, how does the model represent distance from intact forest for regrowth, since distance within the harvested and harvested/burned areas was used in the assessment? Maybe distance from intact forest is not included in ELM-FATES and the authors are just trying to represent the variability in dynamics. The lack of clarity lead me to wonder how important the sampling design was for the overall stufy

Line 270: I found Table 1 and Figure 2 to be quite useful in understanding the sensitivity/uncertainty analysis proposed by the authors.

Line 290: I assume NIR was selected because the results indicate that NIR is most sensitive, correct? If that is the case, perhaps rewording portions of this paragraph to state that the most sensitive spectral band was compared with the ELM-FATES output and save the identification of that band for the results section. At a minimum, stating that NIR is compared because it is most sensitive (referencing results below) is needed.

Lines 344-346: To strengthen the connections between Table 2 and Figure 3, it might help to mark in some way the portion of each time series used to test the sensitivity of the metrics to regrowth.

Figures 4-5: The colors and legend text don't always make sense in these figures. For example, in Figure 6 it looks like the color coding is mixed up for A1 and A2 as they switch colors each year for each time-series. Please check your symbology and the figure captions carefully.

Lines 472-478: This paragraph starts off referring to regrowth, but it appears that the results being discussed are Figure 3a-c, which pertain to the initial disturbance effects. The regrowth is more complicated than that, and is really explored in later paragraphs. This makes me wonder whether it would be better to frame this portion of the discussion in terms of sensitivity of spectral bands for detecting short-term (0-5 year) effects rather than re-growth (6-25 years).

Line 522: Replace "Thought" with "Though"

Lines 555-558 (and elsewhere): By "higher disturbance", do the authors mean higher disturbance magnitude (i.e., tree mortality)? Also, this the comparison meant to highlight a difference between modeled mortality and actual mortality or some sort of comparison between the windthrow and the clearcut. This portion of the paragraph, which is attempting to explain differences between NIR and ELM-FATES results, was confusing to me.

Line 557" Replace "that" with "than"

\*\*Literature Cited\*\*

Alves, DB, RM Llovería, F Pérez-Cabello, L Vlassova. 2018. Fusing Landsat and MODIS data to retrieve multispectral information from fire-affected areas over tropical savannah environments in the Brazilian Amazon. International Journal of Remote Sensing 39: 7919-7941

Bannari, A., D. Morin, F. Bonn, A. R. Huete. 1995. A review of vegetation indices. Remote Sensing Reviews 13:95-120.

Cohen, WB, Z Yang, SP Healey, RE Kennedy, N Gorelick. 2018. A LandTrendr multispectral ensemble for forest disturbance detection. Remote Sensing of Environment 205: 131-140

Foody, GM, DS Boyd, MEJ Cutler. 2003. Predictive relations of tropical forest biomass from Landsat TM data and their transferability between regions. Remote Sensing of Environment 85: 463-474.

Key, CH, NC Benson. 2005. Landscape assessment: remote sensing of severity, the normalized burn ratio. In: Lutes, DC (Ed.), FIREMON: Fire Effects Monitoring, and Inventory System. USDA Forest Service, Rocky Mountain Research, Station, Ogden, UT

McDowell, NG, NC Coops, PSA Beck, JQ Chambers, C Ganodagamage, JA Hicke, C-Y Huang, R Kennedy, DJ Krofcheck, M Litvak, AJH Meddens, J Muss, R Negrón-Juárez, C Peng, AM Schwantes, JJ Swenson, LJ Vernon, AP Williams, C Xu, M Zhao, SW Running, CD Allen. 2015. Global satellite monitoring of climate-induced vegetation disturbances. Trends in Plan Science 20: 114-123

Miller, JD, AE Thode 2007. Quantifying burn severity in a heterogeneous landscape with a relative version of the delta Normalized Burn Ratio (dNBR). Remote Sensing of Environment 109: 66-80

Negrón-Juárez, RI, JA Holm, B Faybishenko, D Magnabosco-Marr, RA Fisher, JK Shuman, AC de Araujo, WJ Riley, JQ Chambers. In review. Landsat NIR band and ELM-FATES sensitivity to forest disturbances and regrowth in the Central Amazon. Biogeosciences Discussion

Schroeder, TA, WB Cohen, Z Yang. 2007. Patterns of forest regrowth following clearcutting in western Oregon as determined from a Landsat time-series. Forest Ecology and Management 243:259-273

---

## Short Comment (SC1) · 3 Feb 2020

This is a very interesting and ambitious paper that links remote sensing and demographic modelling to understand forest disturbance and regrowth in the Amazon. The role of disturbance in forest biomass dynamics and C storage is an important area of research which is challenging to study due to the timescales involved. I think the paper is a valuable contribution but I have some queries about the approach and conclusions.

1. The study is undertaken for one area of the Amazon – are the results (e.g. Fig 3) likely to be extensible across the Amazon, and to other equatorial forests?

2. There are challenges in using LandSat data for tracking forest disturbance and clearance in the Amazon, which lead to biases for smaller magnitude impacts, i.e degradation losses (Milodowksi et al. 2017). These biases are likely to impact the monitoring of forest recovery also. So I suggest extreme caution in interpreting the LandSat time series used here for sensing subtle phenomena like canopy closure and biomass growth. In the results, the statement "The similarity of spectral signatures for the control forests previous to the disturbances suggests comparable structure and species composition" may not be valid. One could equally well conclude that the sensitivity of NIRv is not enough to detect any differences that likely do exist between control old-growth forests. It would help if independent data could show the comparable structure and species composition of the old-growth sites to resolve this issue.

3. The abstract notes that "Statistical methods predict that NIR will return to pre-disturbance values in about 39 years (consistent with observational data of biomass regrowth following windthrows)". I don't find these observational data within the text. It would be very helpful to link the remote sensing directly to ecological time series, so we understand what the NIRv is responding to. I find it hard to understand what "regrowth to old-growth" means in table 3. I think more argumentation is needed to justify the conclusion that "NIR may be used as a proxy in modeling studies aimed at addressing forest regrowth after disturbances." I suggest that more metrics are required to pinpoint 'old-growth' versus 'disturbed' status. Specific ecological metrics would include those that describe biomass stem size distribution, and 3D leaf area density distribution. Li-DAR is an obvious candidate for providing such information.

4. It seems to me that the model simulated quicker LAI recovery and slower biomass recovery to steady state than the remote sensing. The transient response of the model in Fig 7a seems to show overshoot of biomass compared to the 'old-growth' baseline – so when is steady state achieved? LAI (fig 7c) seems to equilibrate (within old growth range) after 15-20 years, much shorter than the NIRv estimate of $\sim$ 40 years. It would be useful to discuss how model transient behaviours can be validated against

independent time series, and how robust the comparisons shown here are.

5. For the evaluation of the FATES model it would help to have direct independent comparison to ecological data. Table 3 could be enhanced with observations for comparison against FATES. It's good to see some model-data comparison to data in fig 9, but how does this size distribution mis-match reflect on the modelling of recovery from disturbance?

---

## Author Comment (AC1) · 7 Feb 2020

Remotely-sensed forest disturbance and change monitoring has been proposed as a basis for improving and constraining vegetation modeling frameworks (McDowell et al. 2015). The current research (Negrón-Juárez et al. in review) proposes and tests components of such a framework by examining the utility of Landsat time series (LTS) data for detecting forest disturbance and recovery dynamics and comparing these results to ELM-FATES predictions for several forest attributes. Using LTS data to identify disturbance is common, but assessing recovery trajectories and timing of recovery is much

less common. The use of LTS data for assessing forest dynamics processes is still a relatively new area of research (Schroeder et al. 2007). More rarely still has been the use of insights into recovery dynamics to assess the capacity of vegetation models to reflect reality and, perhaps, guide model development. Such efforts are particularly challenging in the cloud-covered tropics. The primary conclusions – Landsat-based NIR observations are sensitive to forest regrowth dynamics which compare favorably to model predictions – may prove useful both for the development of monitoring frameworks and for guiding vegetation model testing and development in tropical forests. The paper is well-structured, well-written, and of appropriate length for the material. The citations seems generally appropriate, though I do identify additional literature that should be discussed in the paper (see below). The title and abstract provide a clear and concise description of the work. I do suggest that the authors consider revising the final sentence of the abstract to provide a more impactful conclusion reflecting the potential impact of this work on disturbance and recovery mapping as well as vegetation modeling research. The methods and assumptions seem appropriate, though additional justification of some methods is required (I enumerate those in more detailed comments below). It should be noted that the small sample size of locations used in this study limits the generality of conclusions, but does seem sufficient to assess whether the proposed framework for integrating remote sensing with ELM-FATES is useful. The analysis was straightforward and seems reproducible by other scientists.

Authors: We thank the reviewer for the time and dedication reviewing this manuscript and the comments provided on the use of remote sensing data to improve and develop modeling schemes on disturbance and regrowth. We have included the suggested references and the following:

[Abstract, last sentence] Our results show the potential of Landsat imagery for mapping forest regrowth from disturbance and for validation, improvement, and development of forest regrowth models.

Here I raise several issues that should be addressed to improve the quality and accessibility of the paper.

1. I found the lack of use, or even discussion, of commonly used spectral vegetation indices, such as NBR, NDVI, EVI, or tasseled-cap wetness, greenness, and brightness, to be a significant oversight. Spectral vegetation indices have seen extensive use in remote sensing of terrestrial vegetation (Bannari et al. 1995), including disturbance ecology. For example, NBR is a common basis for disturbance severity mapping (e.g., Key and Benson 2006; Miller and Thode 2007). The signal-to-noise ratio for disturbance mapping in North America tends to be greater for spectral vegetation indices compared to Landsat spectral bands (Cohen et al. 2018). Furthermore, vegetation indices have proven useful in forest biomass mapping (e.g., Foody et al 2003) and forest regrowth monitoring (e.g., Schroeder et al. 2007). I understand that these metric need not form the basis for the current study, but recognizing their application elsewhere and their potential to contribute to future research would be valuable. In particular, some discussion of common spectral vegetation indices that include NIR and/or SWIR1 in the context of the observed sensitivity to regrowth would be valuable.

Response: Vegetation indexes (VI) are based on band combinations before and during disturbance, and forest regrowth. Yet, it is important to understand the band behavior for those conditions (Tucker, 1979). To address this reviewer concern, we have included the following in the revised manuscript:

[Section 4. At the end of last paragraph] In lieu of this development, we show that with successional aging, modeled forest structure returns to pre-disturbed values (through canopy closure) with similar recovery time as NIR, which can be compared against remote sensing metrics like vegetation indices. Nevertheless, the extent to which different vegetation indices (e.g., Normalized Difference Vegetation Index (Rouse et al., 1973), Enhanced Vegetation Index (Huete et al., 2002)) represent the gradient from pre-disturbance through recovery, remains an important area of study.

2. In lines 130-132, it is stated that the burned sites were clearcut, then burned, then
maintained as pasture for a few years before forest regrowth began. I am concerned with referring to cut and then burned areas simply as "burned sites". Fire as a mortality/ disturbance agent has a different impact than fire used as post-disturbance vegetation management. Fire as a disturbance agent within forests will produce various levels of fire severity (e.g., Alves et al. 2018), and thus tree mortality, whereas postdisturbance fire is likely used to benefit forage species and remove woody vegetation, not kill trees in the forest canopy. The latter point is implied by the use of these sites as pasture for several years. It would be more appropriate to refer to these as "cut+burn" or "cut/burn" or something like that. The fire and/or grazing seems to have impacted species composition of the regrowing forest, but attributing the dynamics observed solely to fire seems inappropriate. Furthermore, since no burn simulations were used, is there any reason to have the burned sites without referring to the harvesting history as well? It is likely more appropriate and more interesting to reframe them as cut and burned, which is what they are based on the text. Then, the authors have two types of harvesting, which perhaps provides a more robust assessment of the ELM-FATES model and the complexity of management activities that should be incorporated in the modeling. Even with these concerns, I found the paper to be an interesting attempt to leverage remote sensing in the testing of ecosystem models. Similar applications could lay the groundwork for new developments in Earth system modeling, especially in regions that are traditionally data poor, such as the Amazon.

Response. In the submitted manuscript we defined cleared and burned areas as 'burning areas' for simplicity (line 102). However, for clarity, and in agreement with the reviewer, we will use the whole name: cleared and burned (clear + burn) in the revised manuscript. In our study we emphasized that different disturbances produce different pathways of regrowth that NIR was able to capture.

**Specific and Technical Comments** Line 16: Enumerate which sensors (TM and ETM+) or missions (Landsat 5 and 7) provided the data.

Response. We have included these details in the revised version on Line 16:

[Figure]

[Abstract, second sentence] We used chronosequence of Landsat (Landsat 5 Thematic Mapper and Landsat 7 Enhanced Thematic Mapper Plus) satellite imagery to determine the sensitivity of surface reflectance from all spectral bands ....

Line 42: Forest or successional pathways need to be defined early. Are the authors referring to succession of structural or compositional characteristics?

Response. In the Introduction section, second last sentence of the submitted manuscript we defined successional pathway.

Lines 114-115: Why the different areas? What distinguishes them? I see them defined in lines 211-213. Perhaps the figure caption could reference the distance to edge component of these areas?

Response. For clearcut and burned sites we used areas with different distances from the disturbance edge to determine whether distance is a factor affecting the regrowth. To address this reviewer question, we have included the following in the caption of Figure 1:

[Caption Figure 1] For the pathways of forest regrowth after clearcutting and burning sites we analyzed areas with different distances from the disturbance edge (A1, A2 and A3 in yellow).

Lines 159-162: I think that there may be a few missing words in here.

Response. Yes, we agree. We have revised these lines as follows:

[Section 2.2, first paragraph, second last sentence] We used LEDAPS since has a long time series of data, is promptly available, have high spectral performance comparable to Moderate Resolution Imaging Spectroradiometer (MODIS) (Claverie 160 et al., 2015) and several datasets (Vuolo et al., 2015;Nazeer et al., 2014) and it is suitable for ecological studies in the Amazon (van Doninck and Tuomisto, 2018;Valencia et al., 2016).

Lines 178-179: Was each pixel in the 3x3 pixel box treated as independent (n=9) or averaged (n=1)?

Response. We used the average of the 3x3 pixel box. We have included the following text to clarify this point:

[Section 2.2, third paragraph, second sentence] The spectral characteristics of old-growth forests and their changes after disturbances were investigated using several boxes of 3x3 pixels (Figure 1 b,c,d) as shown in Figures 3a-c. The average of each box was used in our analysis.

Line 179: Figure 3 is cited earlier in the text than figure 2. Consider swapping them or delete the reference to Figure 3 (which I don't think is necessary)

Response. We have remote the citation of Figure 3 in this line.

Lines 193-196: This paragraph seems out of place. As I understand it, the gap-filled data is used in the analysis of the forest regrowth timing described later (lines 213-221). It would be easier if the description of the gap-filling methods appeared just before the analysis of those data.

Response. The methodology of gap filling is very important in the manuscript and was applied to all our analysis of time series of remote sensing data. As suggested by the reviewer, we have placed this paragraph before the analysis of the data in the revised manuscript.

Lines 208-209: I was confused by this sentence. Should it read "The comparison : : : disturbances that was conducted was possible due: : :". Still, I don't agree fully with the sentiment that controlling for those know environmental gradients makes the analysis possible. Perhaps the authors mean that by controlling for these other factors, they make the assessment of forest successional pathways following disturbance more robust.

Response. To address this concern, we have made the following changes:

[Section 2.2, sixth paragraph, first sentence] A direct comparison of successional pathways of forest regrowth from studied disturbances was conducted that is feasible due to the similar conditions of climate, soils, and structure and composition of the old-growth forests.

Lines 209-211: Do the authors mean that greater magnitude wind disturbances tend to have longer recovery times?

Response: We meant that areas with higher disturbance take the longest time to recover.

Lines 254-256: Considering the fact that forest recovery is a major focus on this manuscript, I suggest adding some more detail. Specifically, how does the model represent distance from intact forest for regrowth, since distance within the harvested and harvested/burned areas was used in the assessment? Maybe distance from intact forest is not included in ELM-FATES and the authors are just trying to represent the variability in dynamics. The lack of clarity lead me to wonder how important the sampling design was for the overall study.

Response. We agree with the reviewer that we should provide more of an explanation in the model's representation of disturbed land and distance to intact forest, as this is important for forest recovery. The reviewer's intuition is correct, in that the distance from intact forests was not included in the ELM-FATES simulations used here. A surrounding fully matured forest did not exist around our idealized recovery plot. To help clarify disturbance and recovery dynamics we have added a description to the revised manuscript of seed supply and creation of naturally disturbed patches in ELM-FATES as the forest is recovering from a harvest or windthrow:

[Section 2.3, second paragraph, last sentences] Simulations of disturbance and subsequent vegetation regrowth after disturbance were initiated from this old-growth forest state. The model design used here only allows for simulating intact forests with natural disturbances (e.g., gap dynamics or windthrows) or harvested forests, but not both at

the same time or in adjacent patches. Accounting for distance to intact forests was excluded due to the current limited understanding of seed dispersal mechanisms (i.e., spatial variability, dispersal limitation, etc.) in tropical forests (Terborgh et al., 2019). We use a more general form of seed production, such that the individual cohorts in ELM-FATES use a targeted fraction of net primary production (NPP) during the carbon allocation process (after accounting for tissue turnover and storage demands), which adds to the site-level seed pool for recruitment of new cohorts. Field data was not used to simulate or calibrate the modeled forest recovery post disturbance.

Line 270: I found Table 1 and Figure 2 to be quite useful in understanding the sensitivity/ uncertainty analysis proposed by the authors.

Response. We thank the reviewer for this comment.

Line 290: I assume NIR was selected because the results indicate that NIR is most sensitive, correct? If that is the case, perhaps rewording portions of this paragraph to state that the most sensitive spectral band was compared with the ELM-FATES output and save the identification of that band for the results section. At a minimum, stating that NIR is compared because it is most sensitive (referencing results below) is needed.

Response. We have added the following improvements to address this comment:

[Section 2.3, fourth paragraph, first sentence] In order to evaluate ELM-FATES performance during forest regrowth we compared NIR, the most sensitive band to regrowth (see results), with ELM-FATES outputs of aboveground biomass (AGB, Mg ha-1), total stem density of trees ≥10 cm DBH (stems ha-1), leaf area index (LAI, one-sided green leaf area per unit ground surface area, m2 m-2), and total crown area (m2 m-2) since these variables directly influence the surface reflectance . . .

Lines 344-346: To strengthen the connections between Table 2 and Figure 3, it might help to mark in some way the portion of each time series used to test the sensitivity of the metrics to regrowth.

Response. We have made the following changes to the revised manuscript to address this comment:

Table 2. Test of the significance for the slopes of the time series of six bands from L5 (LEDAPS SR Landsat 5) for the windthrow (period 1991-2011), clearcutting (period 1987-2011), and burning (period 1990-2011) cases in Central Amazonia shown in Figures 3d-f. The critical values (t0.975,8 and t0.975,12 ) for the t distribution were obtained from statistical tables. Bolt represents H1.

Figures 4-5: The colors and legend text don't always make sense in these figures. For example, in Figure 6 it looks like the color coding is mixed up for A1 and A2 as they switch colors each year for each time-series. Please check your symbology and the figure captions carefully.

Response. We noticed a few simple typos in the script plotting the figures. The typos were corrected and figures replotted. There are no changes to our results from these changes. For instance, new Figure 6.

Lines 472-478: This paragraph starts off referring to regrowth, but it appears that the results being discussed are Figure 3a-c, which pertain to the initial disturbance effects. The regrowth is more complicated than that, and is really explored in later paragraphs. This makes me wonder whether it would be better to frame this portion of the discussion in terms of sensitivity of spectral bands for detecting short-term (0-5 year) effects rather than re-growth (6-25 years).

Response. This paragraph refers to the initial changes following the disturbance. To clarify, we have made the following changes: [Section 4, first paragraph, first sentence] Our results show that Landsat reflectance observations were sensitive to the initial changes of vegetation regrowth following windthrows, clearcut, and burning, three common disturbances in the Amazon. Line 522: Replace "Thought" with "Though"

Response. Replaced. Thanks.

Lines 555-558 (and elsewhere): By "higher disturbance", do the authors mean higher disturbance magnitude (i.e., tree mortality)? Also, is the comparison meant to highlight a difference between modeled mortality and actual mortality or some sort of comparison between the windthrow and the clearcut. This portion of the paragraph, which is attempting to explain differences between NIR and ELM-FATES results, was confusing to me.

Response. We agree with the reviewer that "higher disturbance" needs to be more clearly defined. By "higher disturbance" we are referring to either loss of biomass or more openness in the canopy coverage depending on the context of the sentence. We will clarify "higher disturbance" throughout the text. During this section of the discussion we are comparing between the windthrow and the clearcut. We modified these sentences in the revised manuscript:

[Section 4, sixth paragraph, fourth last sentence] ELM-FATES predicted that the timing of peak canopy-coverage was marginally sooner after windthrows compared to clearcuts, opposite to the NIR pattern. This discrepancy may be related to more biomass loss and open canopy coverage, followed by a lack of rapid colonization the higher disturbance in the modeled clearcut.

Line 557" Replace "that" with "than"

Response. Replaced.

**Literature Cited** Alves, DB, RM Llovería, F Pérez-Cabello, L Vlassova. 2018. Fusing Landsat and MODIS data to retrieve multispectral information from fire-affected areas over tropical savannah environments in the Brazilian Amazon. International Journal of Remote Sensing 39: 7919-7941 Bannari, A., D. Morin, F. Bonn, A. R. Huete. 1995. A review of vegetation indices. Remote Sensing Reviews 13:95-120. Cohen, WB, Z Yang, SP Healey, RE Kennedy, N Gorelick. 2018. A LandTrendr multispectral ensemble for forest disturbance detection. Remote Sensing of Environment 205: 131-140 Foody, GM, DS Boyd, MEJ Cutler. 2003. Predictive relations of tropical forest biomass

from Landsat TM data and their transferability between regions. Remote Sensing of Environment 85: 463-474. Key, CH, NC Benson. 2005. Landscape assessment: remote sensing of severity, the normalized burn ratio. In: Lutes, DC (Ed.), FIREMON: Fire Effects Monitoring, and Inventory System. USDA Forest Service, Rocky Mountain Research, Station, Ogden, UT McDowell, NG, NC Coops, PSA Beck, JQ Chambers, C Ganodagamage, JA Hicke, C-Y Huang, R Kennedy, DJ Krofcheck, M Litvak, AJH Meddens, J Muss, R Negrón- Juárez, C Peng, AM Schwantes, JJ Swenson, LJ Vernon, AP Williams, C Xu, M Zhao, SW Running, CD Allen. 2015. Global satellite monitoring of climate-induced vegetation disturbances. Trends in Plan Science 20: 114-123 Miller, JD, AE Thode 2007. Quantifying burn severity in a heterogeneous landscape with a relative version of the delta Normalized Burn Ratio (dNBR). Remote Sensing of Environment 109: 66-80 Negrón-Juárez, RI, JA Holm, B Faybishenko, D Magnabosco-Marr, RA Fisher, JK Shuman, AC de Araujo, WJ Riley, JQ Chambers. In review. Landsat NIR band and ELMFATES sensitivity to forest disturbances and regrowth in the Central Amazon. Biogeosciences Discussion Schroeder, TA, WB Cohen, Z Yang. 2007. Patterns of forest regrowth following clearcutting in western Oregon as determined from a Landsat time-series. Forest Ecology and Management 243:259-273

Huete, A. et al., 2002. Overview of the radiometric and biophysical performance of the MODIS vegetation indices. Remote Sensing of Environment, 83(1-2): 195-213. Rouse, J.W., Hass, R.H., Schell, J.A. and Deering, D.W., 1973. Monitoring vegetation systems in the great plains with ERTS. In: S.C. Freden, E.P. Mercanti and M.A. Becker (Editors), 3rd Earth Resource Technology Satellite (ERTS) Symposium, December 10-14, 1973. GSFC NASA, Washington, DC, USA, pp. 309–317. Terborgh, J., Zhu, K., Loayza, P.A. and Valverde, F.C., 2019. Seed limitation in an Amazonian floodplain forest. Ecology, 100(5). Tucker, C.J., 1979. Red and photographic infrared linear combinations for monitoring vegetation. Remote Sensing of Environment, 8(2): 127-150.

Please also note the supplement to this comment:

[Figure]

https://www.biogeosciences-discuss.net/bg-2019-451/bg-2019-451-AC1-supplement.pdf

[Figure]

[Figure]

Fig. 1.

- L5 NIR $A_T$
- L5 NIR $A_1$
- L5 NIR $A_2$
- L7 NIR $A_T$
- L7 NIR $A_1$
- L7 NIR $A_2$
- median
- Prediction

95% CI of median

Old-growth

Prediction
56.5 years

42.5 years   93 years

---

## Author Comment (AC2) · 28 Feb 2020

Negrón-Juárez et al. Mathew Williams: mat.williams@ed.ac.uk

This is a very interesting and ambitious paper that links remote sensing and demographic modelling to understand forest disturbance and regrowth in the Amazon. The role of disturbance in forest biomass dynamics and C storage is an important area of research which is challenging to study due to the timescales involved. I think the paper is a valuable contribution but I have some queries about the approach and conclusions.

[Figure]

1. The study is undertaken for one area of the Amazon – are the results (e.g. Fig 3) likely to be extensible across the Amazon, and to other equatorial forests?

Response: One of the main reasons in choosing the Central Amazon was due to the available data from windthrows regrowth (that we have published in several manuscripts) and the BDFFP research area, where research on forest regrowth after clearcut and clearcut+burning spans more than 30 years. As mentioned in our discussion "the predominance of Cecropia, after clearcut, and Vismia, after clearcut+burning, have also been found in the Western (Gorchov et al., 1993;Saldarriaga et al., 1986) and the Southern (Rocha et al., 2016) Amazon suggesting that our findings are applicable to other regions. However, an Amazon-wide study is beyond the scope of our work which is to explore the sensitivity of Landsat to most recurrent disturbance types in the Amazon. This is emphasized in our Discussion section, paragraph 5 (last two sentences).

2. There are challenges in using LandSat data for tracking forest disturbance and clearance in the Amazon, which lead to biases for smaller magnitude impacts, i.e degradation losses (Milodowksi et al. 2017). These biases are likely to impact the monitoring of forest recovery also. So I suggest extreme caution in interpreting the LandSat time series used here for sensing subtle phenomena like canopy closure and biomass growth. In the results, the statement "The similarity of spectral signatures for the control forests previous to the disturbances suggests comparable structure and species composition" may not be valid. One could equally well conclude that the sensitivity of NIRv is not enough to detect any differences that likely do exist between control old-growth forests. It would help if independent data could show the comparable structure and species composition of the old-growth sites to resolve this issue.

Response: In order to address these comments we have included (in red color) the following in our manuscript.

[Section 2.2, Paragraph 3, Sentence 5] Landsat is not sensitive to clusters of downed

trees comprising fewer than 8 trees (Negrón-Juárez et al., 2011) or small disturbances (Milodowski et al., 2017). [Section 3.1, First paragraph, last sentence] The similarity of spectral signatures for the control forests previous to the disturbances suggests comparable structure. While Landsat spectral signature alone may not be sensitive to fine differences in species composition, previous research indicated a relatively high floristic similarity between the old-growth forests at our study sites (Negrón-Juárez et al., 2017;De Oliveira and Mori, 1999;Negrón-Juárez et al., 2018;Magnabosco Marra et al., 2018).

3. The abstract notes that "Statistical methods predict that NIR will return to predisturbance values in about 39 years (consistent with observational data of biomass regrowth following windthrows)". I don't find these observational data within the text. It would be very helpful to link the remote sensing directly to ecological time series, so we understand what the NIRv is responding to. I find it hard to understand what "regrowth to old-growth" means in table 3. I think more argumentation is needed to justify the conclusion that "NIR may be used as a proxy in modeling studies aimed at addressing forest regrowth after disturbances." I suggest that more metrics are required to pinpoint 'old-growth' versus 'disturbed' status. Specific ecological metrics would include those that describe biomass stem size distribution, and 3D leaf area density distribution. LiDAR is an obvious candidate for providing such information.

Response: Based on these comments we have made the following changes: [Abstract] Statistical methods predict that NIR will return to pre-disturbance values in about 39 years, a value consistent with our previous observational study of biomass regrowth following windthrows. [Table 3, Title] Time of regrowth to old-growth forest characteristics (years). (In section 3.3 we have explained the term "Regrowth to old-growth": the changes a forest undergoes while it grows from disturbance until it has canopy attributes of an old-growth forest).

[Section 4, paragraph 5] Due to the agreement in recovery timespans for observed NIR trajectories and regrowth, and being the best wavelength for inferring biomass, we suggest that NIR may be used as a proxy in modeling studies aimed at addressing forest regrowth after disturbances. Vegetation characteristics such as tree height, leaf distribution, etc. (currently obtained from Light Detection and Ranging - LiDAR) can improve understanding of regrowth (Almeida et al., 2019), provided the chronosequence of that data encompasses several decades.

4. It seems to me that the model simulated quicker LAI recovery and slower biomass recovery to steady state than the remote sensing. The transient response of the model in Fig 7a seems to show overshoot of biomass compared to the 'old-growth' baseline – so when is steady state achieved? LAI (fig 7c) seems to equilibrate (within old growth range) after 15-20 years, much shorter than the NIRv estimate of âĹij 40 years. It would be useful to discuss how model transient behaviours can be validated against independent time series, and how robust the comparisons shown here are.

Response. The reviewer is correct that the biomass recovery predicted by the model does overshoot the 'old-growth' baseline of AGB. The recovery simulations in ELM-FATES were run for 100 years, and we found AGB reached an equilibrium point starting around simulation year 75 after both disturbances. The biomass equilibrated at 163 MgC ha-1 +/- 1.0 after the clearcut and 163 MgC ha-1 +/- 1.6 after the windthrow, so a little more variance resulted from the windthrow disturbance. We would like to emphasize that AGB is variable across the Central Amazon, and the model stabilizing at ~163 MgC ha-1 is within the observed AGB range (150 MgC ha-1). In section 2.3 of the manuscript, we state that the baseline simulation was spun-up for 400 years and until stable biomass was reached. It was an expected result that LAI would recover and equilibrate faster than NIR (Fig. 7c). We opted to use modeled "canopy-coverage" as a better model comparison to NIR. Canopy-coverage is defined on lines 342 – 343 as the average of crown area, stem density, and LAI since these three variables influence reflectance.

We have updated the text accordingly in Section 3.3:

[Section 3.3, Paragraph 1, Sentence 3] Modeled biomass returned to modeled old-growth forest values quicker after windthrows (37 years, range 21 to 83 years) compared to clearcuts (42 years, range 27 to 80 years). Here the baseline old-growth forest is characterized as when biomass prior to the applied disturbances (ran for 400 years) reached an equilibrium, and with values similar to observed old-growth biomass ($\sim$ 150 MgC ha-1). Interestingly, biomass accumulation from regrowth surpassed the baseline old-growth biomass (108 MgC ha-1), reaching an equilibrium point around 75 years at $\sim$163 MgC ha-1, for both disturbances, and more similar to observed values. The rate of change of biomass regrowth over 50 years switched and was faster in the clearcut simulation (2.5 Mg ha-1 yr-1) than the windthrow simulations (2.0 Mg ha-1 yr-1), which is likely due to the near-zero initial biomass and proportionally greater contribution of fast-growing pioneer species.

5. For the evaluation of the FATES model it would help to have direct independent comparison to ecological data. Table 3 could be enhanced with observations for comparison against FATES. It's good to see some model-data comparison to data in fig 9, but how does this size distribution mis-match reflect on the modelling of recovery from disturbance?

Response: We have compared FATES to independent field data, and have clarified this point in the revised manuscript. In section 3.3 we compare observed biomass (150 MgC ha-1) to the modeled baseline biomass (108 MgC ha-1), and post-disturbance recovery biomass once stable (163 MgC ha-1). In the same section we also compare stem density, stating that FATES simulates low stem density ($\sim$200 stems ha-1) compared to measurements. Since this study is answering hypotheses about remote sensing capabilities over a tropical forest, we think that comparing the model to these measurements and remote sensing observations is adequate. In Section 2.3 we have added text to refer readers to the Holm et al. (2020) study that thoroughly analyzed demography sensitivity and compared a wide range FATES outputs (meteorological, forest attributes, carbon allocation, biomass accumulation) to field data in the Central

Amazon, close to the BDFFP site used here.

We have included the following:

[Section 3.3, Paragraph 3, first sentence] The LAI of the modeled old-growth forest (4.0 m-2 m-2), prior to disturbances, was close to the observed LAI (4.7 m-2 m-2) measured near our site (Chambers et al., 2004). Due to disturbance, the initial modeled LAI (Figure 7c) and total crown area (Figure 7d) decreased, as expected.

References Almeida, D. R. A., Stark, S. C., Schietti, J., Camargo, J. L. C., Amazonas, N. T., Gorgens, E. B., Rosa, D. M., Smith, M. N., Valbuena, R., Saleska, S., Andrade, A., Mesquita, R., Laurance, S. G., Laurance, W. F., Lovejoy, T. E., Broadbent, E. N., Shimabukuro, Y. E., Parker, G. G., Lefsky, M., Silva, C. A., and Brancalion, P. H. S.: Persistent effects of fragmentation on tropical rainforest canopy structure after 20 yr of isolation, Ecological Applications, 29, 10.1002/eap.1952, 2019.

Chambers, J. Q., Tribuzy, E. S., Toledo, L. C., Crispim, B. F., Higuchi, N., dos Santos, J., Araujo, A. C., Kruijt, B., Nobre, A. D., and Trumbore, S. E.: Respiration from a tropical forest ecosystem: Partitioning of sources and low carbon use efficiency, Ecological Applications, 14, S72-S88, 2004.

De Oliveira, A. A., and Mori, S. A.: A central Amazonian terra firme forest. I. High tree species richness on poor soils, Biodiversity and Conservation, 8, 1219-1244, 10.1023/a:1008908615271, 1999.

Gorchov, D. L., Cornejo, F., Ascorra, C., and Jaramillo, M.: THE ROLE OF SEED DISPERSAL IN THE NATURAL REGENERATION OF RAIN-FOREST AFTER STRIP-CUTTING IN THE PERUVIAN AMAZON, Vegetatio, 108, 339-349, 1993.

Holm, J., Knox, R., Zhu, Q., Fisher, R., Koven, C., Lima, A. J. N., Riley, W., Longo, M., Negrón Juárez, R., De Araujo, A. C., Kueppers, L. M., Moorcroft, P., Higuchi, N., and Chambers, J.: The Central Amazon biomass sink under current and future atmospheric CO2: Predictions from big‐leaf and demographic vegetation models,

JGR-Biogeosciences, JGRG21587, 10.1029/2019JG005500, 2020.

Magnabosco Marra, D., Trumbore, S. E., Higuchi, N., Ribeiro, G. H. P. M., Negron-Juarez, R. I., Holzwarth, F., Rifai, S. W., Dos Santos, J., Lima, A. J. N., Kinupp, V. F., Chambers, J. Q., and Wirth, C.: Windthrows control biomass patterns and functional composition of Amazon forests, Global Change Biology, doi:10.1111/gcb.14457, 2018.

Milodowski, D. T., Mitchard, E. T. A., and Williams, M.: Forest loss maps from regional satellite monitoring systematically underestimate deforestation in two rapidly changing parts of the Amazon, Environmental Research Letters, 12, 10.1088/1748-9326/aa7e1e, 2017.

Negrón-Juárez, R. I., Chambers, J. Q., Magnabosco Marra, D., Ribeiro, G. H. P. M., Rifai, S. W., Higuchi, N., and Roberts, D.: Detection of subpixel treefall gaps with Landsat imagery in Central Amazon forests, Remote Sensing of Environment, 115, 3322-3328, 10.1016/j.rse.2011.07.015, 2011.

Negrón-Juárez, R. I., Jenkins, H. S., Raupp, C. F. M., Riley, W. J., Kueppers, L. M., Magnabosco Marra, D., Ribeiro, G., Monteiro, M. T. F., Candido, L. A., Chambers, J. Q., and Higuchi, N.: Windthrow Variability in Central Amazonia, Atmosphere, 8, 10.3390/atmos8020028, 2017.

Negrón-Juárez, R. I., Holm, J. A., Magnabosco Marra, D., Rifai, S. W., Riley, W. J., Chambers, J. Q., Koven, C. D., Knox, R. G., McGroddy, M. E., Di Vittorio, A., Urquiza-Muñoz, J. D., Tello-Espinoza, R., Alegria-Muñoz, W., Ribeiro, G. H. P. M., and Higuchi, N.: Vulnerability of Amazon forests to storm-driven tree mortality, Environmental Research Letters, https://doi.org/10.1088/1748-9326/aabe9f 2018.

Rocha, G. P. E., Vieira, D. L. M., and Simon, M. F.: Fast natural regeneration in abandoned pastures in southern Amazonia, Forest Ecology and Management, 370, 93-101, 10.1016/j.foreco.2016.03.057, 2016.

Saldarriaga, J. G., West, D. C., and Tharp, M. L.: Forest succession in the Upper Rio

Negro of Colombia and Venezuela. , 1986.

---

## Referee Comment (RC2) · Anonymous Referee #2 · 12 May 2020

GENERAL COMMENTS

The paper titled 'Landsat NIR band and ELM-FATES sensitivity to forest disturbances and regrowth in the Central Amazon' examines the use of Landsat satellite data as a tool for quantifying post-disturbance tropical forest recovery following clear-cut logging, burning, and windthrow events in the Central Amazon. The study also compares modeled post-disturbance recovery to the satellite observations, using ELM-FATES (a dynamic global vegetation model), to evaluate whether the model accurately represents differences in forest recovery pathways.

The main claims are as follows:

1. The near infrared (NIR) band provides a useful metric for mapping disturbance events and quantifying the temporal dynamics of post-disturbance recovery.

2. Changes in the NIR band reflect tropical forest succession dynamics following a disturbance event, demonstrated by a decrease in the NIR band that corresponds to the timing of tree loss, a rapid increase in the NIR as tree growth occurs during recovery, followed by a linear decline in NIR back to pre-disturbance conditions over the course of several decades.

3. Clear-cut logging and windthrow events simulated using the version of ELM-FATES in this analysis reproduces Landsat-derived post-disturbance recovery dynamics.

This study offers two valuable contributions:

1. It provides a methodological contribution for identifying how remote sensing data can be used to evaluate demography model performance.

2. It offers an evaluation of ELM-FATES simulated disturbance dynamics and post-disturbance recovery processes following three important disturbance processes.

The study yields interesting results and a useful discussion around the capacity to remotely sense and model tropical forest regrowth following disturbances. The inclusion of spectral leaf reflectance as model output using radiative transfer schemes for direct comparison with remotely sensed data is a welcome idea. I do, however, have several major concerns about the methods and the presentation and interpretation of results, described in detail below. In general, the manuscript would benefit from reorganization and a tighter framing of the narrative. Several paragraphs could be cut down, with unnecessary detail removed, while some descriptions and background information would benefit from greater specificity and detail.

SPECIFIC COMMENTS

- Why use only raw bands? I recognize the importance of understanding band behavior, but as Referee # 1 mentioned, it would be incredibly useful to also look at vegetation

indices (e.g. NDVI, NIRv, EVI) and/or spectrally unmixed bands (e.g. photosynthetic vegetation, non-photosynthetic vegetation, and bare soil) to compare more direct metrics of productivity. Given the amount of non-photosynthetic information in a 30x30 m pixel (e.g. bare soil, branches, etc.), direct comparison of the NIR band to model output like LAI is tricky. NIRv (or EVI, NDVI) is an approach for estimating GPP that will offer a more direct comparison with model output. See: - Badgley, G., Field, C.B. and Berry, J.A., 2017. Canopy near-infrared reflectance and terrestrial photosynthesis. Science advances, 3(3), p.e1602244.

- Why run 20 independent simulations with single PFTs, but no runs with multiple PFTs? It seems highly relevant to look at changes in modeled composition / successional changes to see whether the model qualitatively gets those dynamics right.

L57-58: A quick Google Scholar search reveals several studies using Landsat time-series to map and analyze forest disturbance and recovery dynamics. See, for example:

o Huang, C., Goward, S.N., Masek, J.G., Thomas, N., Zhu, Z. and Vogelmann, J.E., 2010. An automated approach for reconstructing recent forest disturbance history using dense Landsat time series stacks. Remote Sensing of Environment, 114(1), pp.183-198.

o Schroeder, T.A., Wulder, M.A., Healey, S.P. and Moisen, G.G., 2011. Mapping wildfire and clearcut harvest disturbances in boreal forests with Landsat time series data. Remote Sensing of Environment, 115(6), pp.1421-1433.

o Hansen, M.C., Krylov, A., Tyukavina, A., Potapov, P.V., Turubanova, S., Zutta, B., Ifo, S., Margono, B., Stolle, F. and Moore, R., 2016. Humid tropical forest disturbance alerts using Landsat data. Environmental Research Letters, 11(3), p.034008.

o Sen, S., Zipper, C.E., Wynne, R.H. and Donovan, P.F., 2012. Identifying revegetated mines as disturbance/recovery trajectories using an interannual Landsat chronosequence. Photogrammetric Engineering & Remote Sensing, 78(3), pp.223-235.

L84-85: The manuscript would benefit from a more detailed description of the 'range of successional regrowth pathways.' For example, describe what is meant by pathway (recovery of lost/disturbed vegetation to pre-disturbance vegetation), and how pathways could potentially differ (timing, species composition, forest structure, etc.). This will also help clarify how (i) and (ii) differ in L88-89.

L146-163 & L178-191: Much of the information in each of these paragraphs can be tightened.

L188-189: Move L211-213 here so that the different boxes within each site are more clearly linked to the edge effects question. Clarify the distance to edge for each clear cut A1, A2, A3, and burned A1 and A2.

L212-213: clear-cut should read, "selected three areas", while burned site should read, "two areas".

L250-251: It's really too bad that burned area recovery could not be simulated. It would be nice to at least see some discussion of the differences in remotely sensed recovery pathways at all three sites, and how burned area simulations might be expected to differ or not given existing fire models in related DGVMs (e.g. ED), or what aspects of recovery differed at the burn site that should be evaluated in future data-model comparisons.

L316-318: Too much detail for the Results section. Move to Discussion.

L319-320: But L5/L7 data do not reveal anything about species composition. This sentence is misleading.

L480: What is the biophysical motivation/basis for this? Please include a very brief explanation of the relationship.

Clarification requested

**BGD**

L102-103: Provide slightly more descriptive, albeit brief definitions of clear-cut and burned areas. As an example, are clear-cut areas stand-level clearance events where every stem/tree is removed? Is soil compacted by heavy machinery? For burned areas, what is the severity of the fire? Is this typical of fire events in the region? Do all stems/trees burn or is it primarily a brush fire? In addition, please include the complete extent of each disturbance.

L104: Define 'upland' in terms of meters/elevation. Are upland forests characteristic of the region or are lowland (see 50-105 m asl in L136)?

L105: Define 'same geographic region.'

L105: Provide more detail/background information on site characteristics either in the main body of the manuscript or in Supplementary Material. For example, how were the minimal differences in climatic, edaphic, and floristic characteristics determined? What data were used? Provide quantitative comparisons. Additional information on things like AGB, basal area, stem density, etc. will allow the reader to evaluate how similar or different these sites are from one another and how representative they are of the broader landscape.

L134-142: Describe this information for each site separately (e.g. soil characteristics, species diversity/composition, topographic characteristics, mean canopy height, stem density, background mortality rates, etc.).

L113: Why 3x3 windows? Provide an explanation and perhaps compare results using a range of window sizes to evaluate the robustness of results.

L157: Provide very brief explanation of why "especially in tropical forests".

L170-172: Were all Landsat scenes truly cloud free / 0% cloud cover? This seems unlikely. If not, please provide a brief description of what was done to [cloud] mask the data.

L194-196: Show the real data and gap filled data (in Supplementary Material?).

L198 & L341: I am confused about the use of L7 data. Please clarify in the description of the L5 and L7 data precisely when one or both are used.

L205: Briefly explain why these years were selected, e.g. refer to Fig. 3 (d-f).

L220-221: What field observations? What comparisons were made? How was this assessed? Please provide more detail.

L245-250: Aren't there data for the actual sites where analyses were conducted? If so, please provide actual values of mortality, etc. for each site to directly compare the model simulations to the site disturbances.

L294-296: This logic is unclear to me.

L357-361: Are 0.15 and 0.13 mixed up? 0.15 > 0.13. For clarity, it would be useful to compare the relative change in percent reflectance across sites.

o L361: should this read 0.15% yr-1?

o Figs 4-6: These seem to indicate that exactly the same control / old growth values reflectance values were used for each site, although Fig. 1 and earlier descriptions indicate that different control plots were used at each site, which would presumably have different values. Please clarify as this will influence results.

L388-L389: I don't understand how the rate of change can be higher but take longer? Were the starting biomass values different across sites?

L415-417: State this earlier, perhaps on the previous page?

Figure 7: This figure, particularly the AGB panel, seems to imply that the model simulations have not achieved equilibrium after 50 years. Why were simulations cut off at 50 years? How might this impact your results?

L533: Should this read "higher peaks of post-disturbance stem..." instead of "initial stem..."?

Species composition

The changes in species composition at each site is mentioned several times (see L129, L131, L488-490, L492-509). However, it is unclear whether the literature cited to support the differences in pioneer species at each site, and the general changes in composition overlaps directly with the sites included in this study/evaluated using Landsat data.

L488-490: Reword this sentence. This conclusion is overstated based on the data and results reported in the manuscript. Without showing the data on trends in species composition at these sites, this cannot be stated with this much certainty.

o Similarly, L492-509 & L525-528 are all speculation unless you are able to provide the data for these sites. Please clarify that these are speculations or report site-specific data.

Given that changes in species composition provides an important model benchmark, it is unclear why only single PFT simulations were conducted. The manuscript would greatly benefit from additional simulations that include a combination of (at least) early and late successional PFTs.

Timing of disturbances and data availability

L120-132: The different dates associated with each disturbance (1982 – clear-cut, 1984 – burned area, 1987 – windthrow) should be addressed explicitly. Clarify whether analyses (e.g. changes in NIR) are quantified based on recovery since disturbance date or recovery since start of data availability. For example, in Fig. 5 the x-axis title states, "Years since 1984", which is the start date of L5 data, but 2 years after the clear-cut disturbance.

L311-312: Yet you don't have Landsat data immediately following every single disturbance event. Please clarify wording.

- Similarly, if the burned area was used as pastureland until 1987, wouldn't the postdisturbance recovery start data be 1987 instead of 1984 for the burned area site (see L130-131)?

o Fig. 3 highlights the lack of Landsat 5 data for the 1982 clear-cut and 1984 burned area disturbance dates. Given that the L5 launch date was in 1984, there is nothing that can be done about the lack of data prior to 1984. However, I recommend extending the x-axis on Fig. 3 (d-f) back to 1982 to avoid misrepresentation of the data coverage. Including a vertical line at the year of each disturbance in these plots would further clarify this.

L131: Instead of "some" years, could this read 2 or 3 years?

L309: Replace "with" with something like "immediately after" or "within X years of..."

L122: The authors mention in situ data collection on forest structure and species composition since 2011 at the windthrow site. 2011 is well after this forest has recovered. How are these data relevant to this analysis? It is unclear whether they are used directly in analyses in this manuscript. Please clarify.

TECHNICAL CORRECTIONS

L30-32: This statement does not seem fully supported by the results given that observations and model output yielded opposites rate of recovery for clear-cut and windthrow disturbances. What does 'appropriate fidelity' refer to here?

L51: Replace horizontal resolution with spatial resolution

L70: The use of Vegetation Demographic Models (VDMs) as an acronym is unfamiliar. Perhaps replace with Dynamic Vegetation Models (DVMs) or Dynamic Global Vegetation Models (DGVMs).

L120: Include GPS coordinates for the windthrow site, similar to the burned area and clear-cut sites.

L147: ...and Landsat 7 ETM+?

L154: Add 'bands' so that it reads, "L5 bands are derived using. . ."

L159-160: remove "has", "promptly", and "have" so this sentence reads, "We used LEDAPS since a long time series of data is available with high spectral performance. . ."

L168: Insert "dry season" before "months present less cloud cover"?

L173: Mention that all sites are in a single Landsat scene and include the path and row, as is done in the Figure 1 legend.

L178: replace "several boxes" with "n = X boxes."

L179-181: Confusing, reword sentence.

L187: include year – ". . .containing the highest level of SWIR1 in year XXX. . ."

L193: The numbers 27 and 12 don't seem to make sense given the 1984 start of data acquisition to ∼2019.

L202: Insert "in the Manaus region" before "affected by windthrows are dominated. . ."

L213: Insert "Time series of. . ." before "L5 bands were."

L297: Insert "modeled" after "we averaged."

L298: Replace "influence the" with "are more comparable to 30 m. . ."

L301-303: Confusing sentence, reword.

L314: Replace "behavior" with "response."

L330: Clarify at the start of this sentence whether you are referring to all three disturbance types.

L332-333 / Figure 3: Include NIR band values in Fig 3 (d-f) for control plots for direct comparison to emphasize "return to pre-disturbance values".

L338: Replace "become" with "became."

Figure 1: Show inset with all three site locations in the Manaus region together to illustrate their spatial proximity (i.e. a close up of the yellow box in Fig. 1a).

Table 2: Replace "Bolt" with "Bold."

Table 3: Swap the "NIR" and "Model average of forest structure" columns.

---

## Author Comment (AC3) · 1 Jun 2020

Anonymous Referee #2 GENERAL COMMENTS The paper titled 'Landsat NIR band and ELM-FATES sensitivity to forest disturbances and regrowth in the Central Amazon' examines the use of Landsat satellite data as a tool for quantifying post-disturbance tropical forest recovery following clear-cut logging, burning, and windthrow events in the Central Amazon. The study also compares modeled post-disturbance recovery to the satellite observations, using ELM-FATES (a dynamic global vegetation model), to evaluate whether the model accurately represents differences in forest recovery pathways. The main claims are as follows: 1. The near infrared (NIR) band provides

a useful metric for mapping disturbance events and quantifying the temporal dynamics of post-disturbance recovery. 2. Changes in the NIR band reflect tropical forest succession dynamics following a disturbance event, demonstrated by a decrease in the NIR band that corresponds to the timing of tree loss, a rapid increase in the NIR as tree growth occurs during recovery, followed by a linear decline in NIR back to pre-disturbance conditions over the course of several decades. 3. Clear-cut logging and windthrow events simulated using the version of ELM-FATES in this analysis reproduces Landsat-derived post-disturbance recovery dynamics. This study offers two valuable contributions: 1. It provides a methodological contribution for identifying how remote sensing data can be used to evaluate demography model performance. 2. It offers an evaluation of ELM-FATES simulated disturbance dynamics and postdisturbance recovery processes following three important disturbance processes. The study yields interesting results and a useful discussion around the capacity to remotely sense and model tropical forest regrowth following disturbances. The inclusion of spectral leaf reflectance as model output using radiative transfer schemes for direct comparison with remotely sensed data is a welcome idea. I do, however, have several major concerns about the methods and the presentation and interpretation of results, described in detail below. In general, the manuscript would benefit from reorganization and a tighter framing of the narrative. Several paragraphs could be cut down, with unnecessary detail removed, while some descriptions and background information would benefit from greater specificity and detail.

Authors: We thank the reviewer for the comments provided. Our responses to the reviewer comments are in blue and the changes included in the text manuscript are in red.

SPECIFIC COMMENTS 1. Why use only raw bands? I recognize the importance of understanding band behavior, but as Referee # 1 mentioned, it would be incredibly useful to also look at vegetation indices (e.g. NDVI, NIRv, EVI) and/or spectrally unmixed bands (e.g. photosynthetic vegetation, non-photosynthetic vegetation, and bare soil) to

compare more direct metrics of productivity. Given the amount of non-photosynthetic information in a 30x30 m pixel (e.g. bare soil, branches, etc.), direct comparison of the NIR band to model output like LAI is tricky. NIRv (or EVI, NDVI) is an approach for estimating GPP that will offer a more direct comparison with model output. See: - Badgley, G., Field, C.B. and Berry, J.A., 2017. Canopy near-infrared reflectance and terrestrial photosynthesis. Science advances, 3(3), p.e1602244.

Response: Vegetation indexes (VI) and RS metrics are based on band reflectance and therefore it is important to understand the band behavior (Tucker, 1979). For instance the figure bellow shows the NDVI for the windthrows, clearcut and cut+burn sites (we removed the standard deviation at each point in time for clarity). In the figure it can be observed a dynamics of regrowth for each disturbance but does not provide information about that is driving the regrowth.

To address the reviewer concern, we have included this figure as supplementary material. We have included the following in the revised manuscript: [line 579] In lieu of this development, we show that with successional aging, modeled forest structure returns to pre-disturbed values (through canopy closure) with similar recovery time as NIR, which can be compared against remote sensing metrics like vegetation indices (see Supplementary Figure 1). Nevertheless, the extent to which vegetation index (e.g., Normalized Difference Vegetation Index (Rouse et al., 1973), Enhanced Vegetation Index (Huete et al., 2002)) properly represent the successional pathways remains an important area of study.

2. Why run 20 independent simulations with single PFTs, but no runs with multiple PFTs? It seems highly relevant to look at changes in modeled composition / successional changes to see whether the model qualitatively gets those dynamics right.

Response. ELM-FATES is a newly tested demographic model and newly coupled to a land surface model. It was best, at this stage of model application, to understand how tropical forest dynamics play out for each individual PFT type, so investigating solely

early successional recovery behavior or solely late successional behavior, under the same environmental and climate conditions. This will give us a better idea of how to interpret and improve future model results from simulations that combines all PFTs with interacting competition. This is the goal of our next modeling study. We understand your point, and we wanted to reserve parsing out the correlations and dependencies of multiple PFT trait-based competition for other manuscript. Our goal in this manuscript is to determine whether ELM-FATES represent the observed patterns of regrowth.

3. L57-58: A quick Google Scholar search reveals several studies using Landsat time-series to map and analyze forest disturbance and recovery dynamics. See, for example: o Huang, C., Goward, S.N., Masek, J.G., Thomas, N., Zhu, Z. and Vogelmann, J.E., 2010. An automated approach for reconstructing recent forest disturbance history using dense Landsat time series stacks. Remote Sensing of Environment, 114(1), pp.183-198. o Schroeder, T.A., Wulder, M.A., Healey, S.P. and Moisen, G.G., 2011. Mapping wildfire and clearcut harvest disturbances in boreal forests with Landsat time series data. Remote Sensing of Environment, 115(6), pp.1421-1433. o Hansen, M.C., Krylov, A., Tyukavina, A., Potapov, P.V., Turubanova, S., Zutta, B., Ifo, S., Margono, B., Stolle, F. and Moore, R., 2016. Humid tropical forest disturbance alerts using Landsat data. Environmental Research Letters, 11(3), p.034008. o Sen, S., Zipper, C.E., Wynne, R.H. and Donovan, P.F., 2012. Identifying revegetated mines as disturbance/recovery trajectories using an interannual Landsat chronosequence. Photogrammetric Engineering & Remote Sensing, 78(3), pp.223-235.

Response. There are several references on disturbance and pathways of regrowth. It would be impossible to include all of them. We have included those that are relevant to our study of disturbances and pathways of regrowth in tropical forests.

4. L84-85: The manuscript would benefit from a more detailed description of the 'range of successional regrowth pathways.' For example, describe what is meant by pathway (recovery of lost/disturbed vegetation to pre-disturbance vegetation), and how pathways could potentially differ (timing, species composition, forest structure, etc.). This

will also help clarify how (i) and (ii) differ in L88-89.

Response. Pathway is the pattern of vegetation change with time. The process involved is well described in the Introduction section, first paragraph, second last sentence.

We have included the following:

[line 42] In general, it is known that forest pathways of regrowth (the pattern of regrowth) initiate with fast-growing and shade-intolerant species (pioneers) that establish from seeds and dominate a few years after disturbance, followed by recruitment and establishment of shade-tolerant species, and finally a closed-canopy old growth forest (Chazdon, 2014;Denslow, 1980;Mesquita et al., 2001;Swaine and Whitmore, 1988).

5. L146-163 & L178-191: Much of the information in each of these paragraphs can be tightened.

Response. We believe that is appropriated to provide those detail.

6. L188-189: Move L211-213 here so that the different boxes within each site are more clearly linked to the edge effects question. Clarify the distance to edge for each clear cut A1, A2, A3, and burned A1 and A2.

Response. We have done the changed suggested by the reviewer.

7. L212-213: clear-cut should read, "selected three areas", while burned site should read, "two areas".

Response. As indicated in the same lines there are four areas for clearcut ( A1, A2, A3, and AT) and three areas for cut+burn (A1, A2, and AT)

8. L250-251: It's really too bad that burned area recovery could not be simulated. It would be nice to at least see some discussion of the differences in remotely sensed recovery pathways at all three sites, and how burned area simulations might be expected to differ or not given existing fire models in related DGVMs (e.g. ED), or what aspects of recovery differed at the burn site that should be evaluated in future data-model comparisons.

We understand that it's unfortunate there are not simulations of recovery from burned areas. The fire model is not yet completed or fully tested in the ELM-FATES model. Jacquelyn Shuman (co-author) is developing and finishing the ELM-FATES fire model and has been working diligently on fire modeling and multiple research questions. ELM-FATES uses a modification of the SPITFIRE module from (Thonicke et al., 2010), and development has required adaptation of SPITFIRE for the patch framework of FATES. We did not want to put an unfinished version of recovery after fire in this manuscript prior to that. We will make this clearer in the revision.

[Section 2.3, 2nd paragraph, 6th sentence]: The fire module in ELM-FATES is currently under final development and testing and therefore burned simulations are not included in this study."

9. L316-318: Too much detail for the Results section. Move to Discussion.

Response. We have moved the sentence to discussion. We have done the following change.

[line 472] Our results show that Landsat reflectance observations were sensitive to the initial changes of vegetation following windthrows, clearcut, and cut+burn, three common disturbances in the Amazon. Specifically, a decrease in NIR and an increase in SWIR1 were the predominant spectral changes immediately (within a few years) following disturbances. The increase in SWIR1 was different among the disturbances with the maximum increase observed in the cut+burn, followed by clearcut and then the windthrow site. The highest increase in SWIR1 in cut+burn sites may be related to the highest thermal emission of burned vegetation (Riebeek, 2014). Likewise, the relatively higher moisture content of woody material in the windthrow site decreases the reflection of SWIR2. On the other hand, in our control (old-growth) forests, we observed typically high NIR reflectance due to the cellular structure of leaves (Chapter

7 in Adams and Gillespie, 2006), absorption of red radiation by chlorophyll (Tucker, 1979), and absorption of SWIR1 by the water content in leaves (Chapter 7 in Adams and Gillespie, 2006).

10. L319-320: But L5/L7 data do not reveal anything about species composition. This sentence is misleading.

Response. Our control forest are located in old-growth forests. Due to edaphic and climate similarities it is very likely that the spectral similarities is related to comparable structure and floristic composition. We have done the following change.

[line 319] The similarity of spectral signatures for the control forests previous to the disturbances suggests comparable structure and floristic composition.

11. L480: What is the biophysical motivation/basis for this? Please include a very brief explanation of the relationship.

Response: Due to lack of leaf SWR1 increase and details are included in Section 4, 1st paragraph.

Clarification requested 12. L102-103: Provide slightly more descriptive, albeit brief definitions of clear-cut and burned areas. As an example, are clear-cut areas stand-level clearance events where every stem/tree is removed? Is soil compacted by heavy machinery? For burned areas, what is the severity of the fire? Is this typical of fire events in the region? Do all stems/trees burn or is it primarily a brush fire? In addition, please include the complete extent of each disturbance.

Response. We have redefine the term burned as cut+burn in all the manuscript. Soil compacts easily so machinery was avoided the BDFFP (Lessons from Amazonia: The Ecology and Conservation of a Fragmented Forest, Chapter 4, The Biological Dynamics of Forest Fragment Project). We have included this reference in the manuscript. Areas of disturbances are shown in Figure 1. Windthrows, clearcut and cut+burn are typical in the region as mentioned in the manuscript. Further details are in the second

paragraph of section 2.1

We have included the following changes in the revised manuscript:

[line 100] Forests in the Central Amazon affected by windthrow (Figure 1), clearcut, and cut+burn were addressed in this study. In clearcut areas, forests are cut and cleared and in cut+burn areas forest are cleared and burned (Mesquita et al., 2001;Mesquita et al., 2015;Lovejoy and Bierregaard, 1990).

[line 125] The BDFFP was established and managed in early 1980's by Brazil's National Institute for Research in Amazonia (INPA) and the Smithsonian Institution, and is the longest running experiment of forest fragmentation in the tropics (Bierregaard et al., 1992;Lovejoy et al., 1986;Laurance et al., 2011;Tollefson, 2013;Laurance et al., 2018). Further details of the BDFFP are in Bierregaard et al. (2001).

13. L104: Define 'upland' in terms of meters/elevation. Are upland forests characteristic of the region or are lowland (see 50-105 m asl in L136)? L105: Define 'same geographic region.' L105: Provide more detail/background information on site characteristics either in the main body of the manuscript or in Supplementary Material. For example, how were the minimal differences in climatic, edaphic, and floristic characteristics determined? What data were used? Provide quantitative comparisons. Additional information on things like AGB, basal area, stem density, etc. will allow the reader to evaluate how similar or different these sites are from one another and how representative they are of the broader landscape. L134-142: Describe this information for each site separately (e.g. soil characteristics, species diversity/composition, topographic characteristics, mean canopy height, stem density, background mortality rates, etc.).

Response. We have integrated these four comments since they occur in the same paragraph. Upland refers to no flooding. We have change geographical region to region. Region refers to a land area that has common features. This features are mentioned in the same sentence. The last paragraph in Section 2.1 describe these features

with proper references for readers that would like a detailed description. Following the reviewer suggestion we have included the basal area for old-growth forest trees with DBH ïĆş 10 cm that for the BDFFP was assessed by Laurence et al. 2010 and for the Tumbira windthrow was published in the PhD thesis of Daniel Magnabosco Marra and for other windthrows in the area by the same author.

We have done the following changes:

[line 103] The windthrow, clearcut, and cut+burn sites used in this study were selected based on the following conditions: (a) prior to disturbance they were upland (no flooding) old-growth forest and located in the same region, with similar climatic, edaphic, and floristic differences; (b) long-term records of satellite imagery and corresponding field data before and after disturbance are available; and (c) no subsequent disturbance has occurred.

[line 134] In the Manaus region the mean annual temperature is 27°C (with higher temperatures from August to November, and peak in October) and the mean annual rainfall is 2,365 mm with the dry season (rainfall < 100 mm month-1 (Sombroek, 2001)) from July to September (Negrón-Juárez et al., 2017). The topography is relatively flat with landforms ranging from 50-105 m above sea level (Laurance et al., 2011;Renno et al., 2008;Laurance et al., 2007), and the mean canopy height is $\sim$ 30 m, with emergent trees reaching 55 m (Laurance et al., 2011;Lima et al., 2007;Da Silva, 2007). The soil in this region are ferrosols (Quesada et al., 2011;Bierregaard et al., 2001;Ferraz et al., 1998) (Food and Agriculture Organization FAO classification) and with similar floristic composition (Bierregaard et al., 2001;Carneiro et al., 2005;Vieira et al., 2004;Higuchi et al., 2004). In the BDFFP, and for old-growth forest trees with DBH $\geq$ 10 cm, there are 261ïĆś18 species per hectare, the stem density is 608 ïĆś 52 stems ha$-1$ and the basal area is $\sim$ 28 m2 ha-1 (Laurance et al., 2010). These values are representative of the region (da Silva et al., 2002;Vieira et al., 2004;Carneiro et al., 2005;Magnabosco Marra et al., 2014;Magnabosco Marra et al., 2018;Magnabosco Marra, 2016). In this region 93% of stems are between 10 and 40 cm in DBH (Higuchi et al., 2012) and the

annual tree mortality is of 8.7 trees ha−1 for trees ≥ 10 cm in DBH (Higuchi et al., 1997).

14. L113: Why 3x3 windows? Provide an explanation and perhaps compare results using a range of window sizes to evaluate the robustness of results.

Response. Windows of these size capture better the spectral signature of events under study. The standard deviation from these windows is small, corroborating their use. These windows are also based on our experience in this type of analysis.

15. L157: Provide very brief explanation of why "especially in tropical forests".

Response. Due to high atmospheric effects, tropical forest show higher differences in surface reflectance between L5 and L7.

We have included the following in the manuscript:

[line 156 ] Though L5 and L7 use the same wavelength bands they are different sensors and differences in surface reflectance may exist, especially in tropical forests due to high atmospheric effects (Claverie et al., 2015).

16. L170-172: Were all Landsat scenes truly cloud free / 0% cloud cover? This seems unlikely. If not, please provide a brief description of what was done to [cloud] mask the data.

Response. As explained in the manuscript: Only images with cloud free, cloud shadow free, and haze free over our disturbed areas were used to eliminate errors associated with these elements. For this procedure, visual inspection of visible bands and quality information from L5 and L7 were used. The following lines also mentioned the dates of the images: The dates of L5 images used were (Landsat 5 operational imaging ended in 2011) 6/1/1984, 7/6/1985, 7/12/1987, 8/2/1989, 7/20/1990, 8/8/1991, 7/31/1994, 6/21/1997, 7/26/1998, 7/13/1999, 7/24/2003, 8/4/2007, 8/6/2008, 7/27/2010, and 8/31/2011.

17. L194-196: Show the real data and gap filled data (in Supplementary Material?). Response. It is shown in Figure 3.

18. L198 & L341: I am confused about the use of L7 data. Please clarify in the description of the L5 and L7 data precisely when one or both are used.

Response. As mentioned in the manuscript (lines 155-156), L7 images were used to corroborated our predictions.

19. L205: Briefly explain why these years were selected, e.g. refer to Fig. 3 (d-f).

Response. It was described in lines 119-132.

20. L220-221: What field observations? What comparisons were made? How was this assessed? Please provide more detail.

Response. Section 2.1 contains this information. For accuracy we have include the following changes:

[lines 219] The predictions were compared with published field observations (Section 2.1) where data were available and L7 images were used to assess the reliability of our predictions.

21. L245-250: Aren't there data for the actual sites where analyses were conducted? If so, please provide actual values of mortality, etc. for each site to directly compare the model simulations to the site disturbances.

Response. Such data is provided in the same lines with proper mention of the references.

22. L294-296: This logic is unclear to me. When benchmarking a model against observations, it's usually a good practice to evaluate multiple model outputs. Therefore, we wanted to report the model outputs of multiple forest variables. With respect to observational data we concluded that biomass, stem density, and tree crowns with live foliage were appropriate model results to explore. Further logic behind this approach

is that multiple forest characteristics can contribute to NIR reflectance, and NIR is not necessarily directly tied to one variable (Ollinger, 2011). NIR can change based on the density of vegetation, biomass levels, and amount of tree crown that is live. This comparison to NIR is another reason we chose to explore multiple model outputs. We have introduced the following change in the manuscript: [line 293] We suggest that testing an array of modeled forest variables provides a robust approach for comparison to NIR, due to multiple forest characteristics contributing to and affecting NIR reflectance (Ollinger 2010), and reduces model unknowns and biases that can arise when using only one model variable. 23. L357-361: Are 0.15 and 0.13 mixed up? 0.15 > 0.13. For clarity, it would be useful to compare the relative change in percent reflectance across sites.

Response. In windrown areas NIR decrease as 0.13% y-1 but then double (0.26% y-1). In clearcut the decrease is 0.4% y-1. For the cut+burn site the decrease is 0.15% y-1. The decrease of NIR is lower in the cut+burn site. For clarity we have include the following in our manuscript

[line 357] During the first 12 years following the windthrow, the spline curve fitted to the NIR data decreased by ∼0.13% y-1 after which the rate of decrease doubled (0.26% y-1, Figure 4).

24. L361: should this read 0.15% yr-1?

Response. Yes. It is 0.15% y-1. Thanks for the correction. We have included it in the manuscript.

25. Figs 4-6: These seem to indicate that exactly the same control / old growth values reflectance values were used for each site, although Fig. 1 and earlier descriptions indicate that different control plots were used at each site, which would presumably have different values. Please clarify as this will influence results.

Response. As mentioned in Section 3.2, 2nd paragraph we used the average NIR and

the variability from all old-growth forest sites is shown in the gray bar in Figures 4-6.

26. L388-L389: I don't understand how the rate of change can be higher but take longer? Were the starting biomass values different across sites?

Response. Your assumption is correct about different starting biomass values between the two sites (windthrow and clearcut). This can be seen in Figure 7a, where the biomass for clearcut starts at near-zero, and the biomass for windthrows starts around 30 MgC ha-1. We have also updated the sentence to make this point clearer. [line 390] which was due to the clearcut site recovering from initial biomass of near-zero and proportionally greater contribution of fast-growing pioneer species."

27. L415-417: State this earlier, perhaps on the previous page?

Response. We are comparing modeled changes in LAI and the NIR. We believe that as organized, the sentences provide a better flow.

28. Figure 7: This figure, particularly the AGB panel, seems to imply that the model simulations have not achieved equilibrium after 50 years. Why were simulations cut off at 50 years? How might this impact your results?

Response. We were mainly interested in determining how long the simulations took to recover to pre-disturbance levels so that the simulations could be compared to the remote sensing reflectance, and when the reflectance returned to old-growth forest values. We did happen to run simulations out to 100 years to observe longer patterns in regrowth. The simulated biomass, after both disturbances, did reach an equilibrium around ∼90 years.

29. L533: Should this read "higher peaks of post-disturbance stem: : :" instead of "initial stem: : :"? Response. Yes. We have rewritten the sentence as: [line 532] The strongest agreement, which can be used for future benchmarking, occurred because ELM-FATES predicted higher peaks of post-disturbance stem density and LAI in clearcuts than in windthrows, consistent with the higher peak of NIR from clearcuts

(Figure 5 vs. Figure 4).

Species composition

30. The changes in species composition at each site is mentioned several times (see L129, L131, L488-490, L492-509). However, it is unclear whether the literature cited to support the differences in pioneer species at each site, and the general changes in composition overlaps directly with the sites included in this study/evaluated using Landsat data.

Response. After clearcut the area is dominated by Cecropia and after cut+bur by Vismia. This is described in section 2.1. We have also included that in windthrows our published observational studied shown that Cecropia is a dominant pioneer. We have done the following change in the manuscript:

[line 121] At this site, data on forest regrowth including forest structure and species composition for trees ≥10 cm DBH (diameter at breast height, 1.3 m) were collected since 2011 covering disturbed (dominated by the genus Cecropia) and undisturbed areas (Magnabosco Marra et al., 2018).

31. L488-490: Reword this sentence. This conclusion is overstated based on the data and results reported in the manuscript. Without showing the data on trends in species composition at these sites, this cannot be stated with this much certainty. o Similarly, L492-509 & L525-528 are all speculation unless you are able to provide the data for these sites. Please clarify that these are speculations or report site-specific data.

Response. In the Discussion we are placing our result in context relative to previous studies that in turn (the previous studies) were properly referenced.

32. Given that changes in species composition provides an important model bench-mark, it is unclear why only single PFT simulations were conducted. The manuscript would greatly benefit from additional simulations that include a combination of (at least) early and late successional PFTs.

We decided that it would be best, at this stage of model application, to understand how tropical forest dynamics play out for each individual PFT type, so investigating solely early successional recovery behavior or solely late successional behavior, under the same environmental and climate conditions. We understand your concern, however we wanted to reserve parsing out the many correlations and dependencies of multiple PFT trait-based competition for other manuscript. The ELM-FATES model is continually being developed, and it was decided that including combined interactions of many PFTs, and evaluating any changes to composition, wouldn't lead to robust results at this time.

Timing of disturbances and data availability

33. L120-132: The different dates associated with each disturbance (1982 – clear-cut, 1984 – burned area, 1987 – windthrow) should be addressed explicitly. Clarify whether analyses (e.g. changes in NIR) are quantified based on recovery since disturbance date or recovery since start of data availability. For example, in Fig. 5 the x-axis title states, "Years since 1984", which is the start date of L5 data, but 2 years after the clear-cut disturbance.

Response. We thank the reviewer for this observations. Due to cloud cover over the windthrown area there is no data in 1984 for this disturbance. We have included the following clarification.

[line 174]: The dates of L5 images used were (Landsat 5 operational imaging ended in 2011) 6/1/1984 (except for the windthrow), 7/6/1985, 7/12/1987, 8/2/1989, 7/20/1990, 8/8/1991, 7/31/1994, 6/21/1997, 7/26/1998, 7/13/1999, 7/24/2003, 8/4/2007, 8/6/2008, 7/27/2010, and 8/31/2011. The dates of L7 images used were 8/7/2011, 6/22/2012, 6/12/2014, 8/2/2015 and 8/7/2017.

34. L311-312: Yet you don't have Landsat data immediately following every single disturbance event. Please clarify wording.
Response. We are describing here a process observed for windthrows. Also, thought the spectral response to fires encompasses several references we have decided to use one reference that is a review. We have included the following:

[line 310] This decrease in NIR was due to exposed woody material and dry leaves (typical after windthrow and clearcutting) that have been observed in windthrows (Negrón-Juárez et al., 2010a; Negrón-Juárez et al., 2011) or the dark surface following burning (Pereira et al. 1997). For windthrows, such effects last about one year after which vegetation regrowth covers the ground surface (Negrón-Juárez et al., 2010a;Negrón-Juárez et al., 2011).

35. Similarly, if the burned area was used as pastureland until 1987, wouldn't the post-disturbance recovery start data be 1987 instead of 1984 for the burned area site (see L130-131)?

Response: Yes, and this is specified in line 205 of the manuscript.

36. Fig. 3 highlights the lack of Landsat 5 data for the 1982 clear-cut and 1984 burned area disturbance dates. Given that the L5 launch date was in 1984, there is nothing that can be done about the lack of data prior to 1984. However, I recommend extending the x-axis on Fig. 3 (d-f) back to 1982 to avoid misrepresentation of the data coverage. Including a vertical line at the year of each disturbance in these plots would further clarify this.

Response. We have included the reviewer suggestion. The new Figure 3 is :

Figure 3: L5 (LEDAPS SR Landsat 5) spectral characteristics for (a) windthrow (July 12, 1987), (b) clearcut (June 1 , 1984), and (c) cut+burn (July 12, 1987) (in red) and control (old-growth) forests (in green) sites. Time series of each L5 spectral bands for (d) windthrow, (e) clearcut, and (f) cut+burn sites. The bars represent the standard deviation from all pixels from all 3ïĆť3 boxes comprising the respective disturbances showed in Fig. 1. Vertical dashed lines represent the year of the disturbance.

[Figure]

37. L131: Instead of "some" years, could this read 2 or 3 years?

Response. We have included the reviewer suggestion:

[line 130] The cut+burn site is located in the Dimona farm (Figure 1d), which was clearcut and burned in September 1984 and maintained as pasture for 2 or 3 years and then abandoned. By 1993 this site was 6 years old and dominated by the pioneer tree genus Vismia (Mesquita et al., 1999;Mesquita et al., 2001).

38. L309: Replace "with" with something like "immediately after" or "within X years of:
: :"

Response. We have included the reviewer suggestion.

Overall, all L5 bands showed an increase in surface reflectance immediately after windthrow, clearcut, and burned sites e

39. L122: The authors mention in situ data collection on forest structure and species composition since 2011 at the windthrow site. 2011 is well after this forest has recovered. How are these data relevant to this analysis? It is unclear whether they are used directly in analyses in this manuscript. Please clarify.

Response. Based on this data we assessed the time of recovery of windthrows that is later used to compare with our RS results that is described in our Discussion. Provide these details is important since allow a comparison of our RS data with observations.

TECHNICAL CORRECTIONS

40. L30-32: This statement does not seem fully supported by the results given that observations and model output yielded opposites rate of recovery for clear-cut and windthrow disturbances. What does 'appropriate fidelity' refer to here?

Response. We have modified or sentence as:

The similarity of ELM-FATES predictions of regrowth patterns after windthrow and

clearcut to those of the NIR results suggest that the patterns of forest regrowth for these disturbances are well represented within ELM-FATES.

41. L51: Replace horizontal resolution with spatial resolution

Response. We have included the reviewer suggestion.

42. L70: The use of Vegetation Demographic Models (VDMs) as an acronym is unfamiliar. Perhaps replace with Dynamic Vegetation Models (DVMs) or Dynamic Global Vegetation Models (DGVMs).

Response. VDM is a new acronym that is being introduced to the vegetation modeling community that is based on the Fisher et al. (2018) review manuscript ("Vegetation Demographics in Earth System Models: A review of progress and priorities"). The term VDM is also being used in all current and future papers including FATES, and we wanted to be consistent. We are choosing to adopt this acronym, and its slightly updated definition of including "demographic". FATES is not classified as a DGVM, because first generation DGVMs do not capture many demographic processes considered important for predicting ecosystem composition and function, including canopy gap formation, vertical light competition, competitive exclusion, and successional recovery from disturbance. 43. L120: Include GPS coordinates for the windthrow site, similar to the burned area and clear-cut sites.

Response. We have included the coordinates

44. L147: : : :and Landsat 7 ETM+? Response. It is mentioned in the following sentences in the same paragraph. 45. L154: Add 'bands' so that it reads, "L5 bands are derived using: : :"

Response. Bands and subsequent changes suggested were included.

46. L159-160: remove "has", "promptly", and "have" so this sentence reads, "We used LEDAPS since a long time series of data is available with high spectral performance: : :"

Response. The suggested change was introduced in the manuscript.

47. L168: Insert "dry season" before "months present less cloud cover"?

Response. We included the change suggested by the reviewer.

48. L173: Mention that all sites are in a single Landsat scene and include the path and row, as is done in the Figure 1 legend.

Response. The suggested addition was included in the respective sentence.

49. L178: replace "several boxes" with "n = X boxes."

Response. We included the suggested change.

50. L179-181: Confusing, reword sentence.

Response. We have done the following change:

[line 178] Spectral characteristics for old-growth forest for each site were determined from boxes located in the same position of the disturbance but previous to disturbance and/or from adjacent areas.

51. L187: include year – ": : :containing the highest level of SWIR1 in year XXX: : :"

Response. The year 1987 was included in the sentence

52. L193: The numbers 27 and 12 don't seem to make sense given the 1984 start of data acquisition to _2019.

Response. In line 173 of the submitted manuscript we mentioned that Landsat 5 stop its operation in 2011, 28 years of data since 1984. In that period we got 15 images to use over our study areas listed in lines 174 and 175.

We have included the following changes in the manuscript:

[Line 193 ] L5 data for the windthrow, clearcut, and cut+burn sites encompass a period of 28 years with 13 years of missing data

53. L202: Insert "in the Manaus region" before "affected by windthrows are dominated: : :"

Response. We inserted the words suggested by the reviewer.

54. L213: Insert "Time series of: : :" before "L5 bands were."

Response. Done

55. L297: Insert "modeled" after "we averaged."

Response. Done

56. L298: Replace "influence the" with "are more comparable to 30 m: : :"

Response. We have done the following change:

[line 297] In addition, we averaged modeled outputs of crown area, stem density, and LAI since each of these variables influence the reflectance of forests, and defining this average as the modeled 'canopy-coverage'

57. L301-303: Confusing sentence, reword.

Response. The sentence was modified as:

Modeled diameter growth (cm y-1) for trees with DBH $\geq$10 cm is also show to provide information of the successional dynamics of forest stands within ELM-FATES.

58. L314: Replace "behavior" with "response."

Response. Done.

59. L330: Clarify at the start of this sentence whether you are referring to all three disturbance types.

Response. We have done the following change:

About six years after the disturbances,

60. L332-333 / Figure 3: Include NIR band values in Fig 3 (d-f) for control plots for direct comparison to emphasize "return to pre-disturbance values".

Response. Figures 4-6 show exactly this comparison.

61. L338: Replace "become" with "became." Response. Done.

Figure 1: Show inset with all three site locations in the Manaus region together to illustrate their spatial proximity (i.e. a close up of the yellow box in Fig. 1a).

62. Response. Based on the reviewer suggestion the new Figure 1 is:

63. Table 2: Replace "Bolt" with "Bold."

Response. Done. Thanks.

64. Table 3: Swap the "NIR" and "Model average of forest structure" columns. Response. Done. Thanks.

[revised manuscript text omitted]
;2, 2001. Thonicke, K., Spessa, A., Prentice, I. C., Harrison, S. P., Dong, L., and Carmona-Moreno, C.: The influence of vegetation, fire spread and fire behaviour on biomass burning and trace gas emissions: results from a process-based model, Biogeosciences, 7, 1991-2011, 10.5194/bg-7-1991-2010, 2010. Tollefson, J.: SPLINTERS OF THE AMAZON, Nature, 496, 286-289, 2013. Tucker, C. J.: Red and photographic infrared linear combinations for monitoring vegetation, Remote Sensing of Environment, 8, 127-150, 10.1016/0034-4257(79)90013-0, 1979. Vieira, S., de Camargo, P. B., Selhorst, D., da Silva, R., Hutyra, L., Chambers, J. Q., Brown, I. F.,

[Figure]

[Figure]

Higuchi, N., dos Santos, J., Wofsy, S. C., Trumbore, S. E., and Martinelli, L. A.: Forest structure and carbon dynamics in Amazonian tropical rain forests, Oecologia, 140, 468-479, 10.1007/s00442-004-1598-z, 2004.

Please also note the supplement to this comment:
https://www.biogeosciences-discuss.net/bg-2019-451/bg-2019-451-AC3-supplement.pdf

———————————————————

[Figure]

**Fig. 1.** Fig 1. Supp Fig 1

**(a) windthrow**

old-growth forest
disturbance

**(b) clearcut**

**(c) cut+burn**

**(d) windthrow**

BLUE   GREEN   RED
NIR     SWIR1    SWIR2

**(e) clearcut**

**(f) cut+burn**

**Fig. 2.** New Figure 3

(a)

(d)cut+burn · ·(c)clearcut

(b)windthrow

100 km

Google Earth

2200 km

(b)

3°S

3°1'30"S

2km

60°46'30"W    60°45'W    60°43'30"W

(c)

2°21'S

A₁

A₂

A₃

2°22'30"S

2km

59°57'W

(d)

2°19'30"S

A₂

A₁

2°21'S

2km

60°7'30"W    60°6'W

**Fig. 3.** New Figure 1